

# Stable sulfur isotope measurements to trace the fate of $SO_2$ in the Athabasca oil sands region

Neda Amiri[1], Ann-Lise Norman[1], Roya Ghahreman[2], Ofelia Rempillo[3], Hans D. Osthoff[4],
Charles A. Odame-Ankrah[5], and Travis Tokarek[4]

[1]Department of Physics and Astronomy, University of Calgary, Calgary ,Canada
[2]Environment and Climate Change Canada, Toronto, Canada
[3]Physics Department,De La Salle University, Manila Philippines
[4]Department of Chemistry, University of Calgary, Calgary, Canada
[5]Global Analyzer Systems Ltd., 1411 25th Ave NE, Calgary, AB. T2E7L6

*Correspondence to:* Ann-Lise Norman (alnorman@ucalgary.ca)

**Abstract.** The concentration and sulfur isotopic composition of $SO_2$ and size segregated sulfate aerosols were determined for Air Monitoring Station 13 (AMS13) at Fort MacKay in the Athabasca oil sands region, northeastern Alberta, Canada as part of the Joint Canada-Alberta Implementation Plan for Oil Sands Monitoring (JOSM) campaign from Aug 13 to Sep 5, 2013. Sulfate aerosols were collected on filters by a high volume sampler, with 12 or 24 hour time intervals.

Significant positive correlations between $SO_2$ to sulfate conversion ratio (F(s)) and the concentration of $\alpha$-pinene (r=0.85), $\beta$-pinene (r=0.87), and limonene (r=0.82) during daytime and with other alkenes and aromatics (tetrachloroethene (r=0.69), 1-methyl-4benzene (r=0.71) and Ethenylbenzene (r=0.66)) indicate that $SO_2$ oxidation by Criegee intermediates can be a potential oxidation pathway in highly polluted regions.

Enriched $^{34}S$ sulfur, largely as $SO_2$, was emitted by a nearby Chemical Ionization Mass Spectrometer (CIMS) and affected
isotope samples for a portion of the sampling period. When it was realized this could be a useful tracer, samples collected were broken into two sets. The first set includes periods when the CIMS was not running (CIMS-OFF) and no enriched $^{34}S$ sulfur was emitted. The second set is for periods when the CIMS was running when sampling (CIMS-ON) and $^{34}S$ sulfur values expected to affect $SO_2$ and sulfate samples.

$\delta^{34}S$ values for sulfate aerosols with $D > 0.49\mu$m during CIMS-OFF periods (no tracer $^{34}SO_2$ present) indicating the sulfur
isotope characteristics of sulfate in the region, were isotopically lighter (down to -4.5 ‰) than what was expected according to potential sulfur sources in the Athabasca oil sands region (+3.9 ‰ to +11.5‰). For sulfate aerosols with $D < 0.49\mu$m, $\delta^{34}S$ values were lighter than $\delta^{34}S$ value for $SO_2$ measured for the same time interval. A significant anti correlation between $\delta^{34}S$ values of sulfate aerosols with $D < 0.49\mu$m (mostly secondary) and the concentration of Fe and Mn (r=0.80 and r=0.76 respectively) was observed. These pieces of information indicate that $SO_2$ was likely oxidized by a transition metal catalyzed
pathway on the surface of aerosols. This process involved $O_2$ and $Fe^{3+}$ or $Mn^{2+}$ (Transition Metal Ion (TMI)-catalyzed pathway) and this is the only known oxidation pathway favoring lighter sulfur isotopes. Analysis of $^{34}S$ enhancements of sulfur compounds during CIMS-ON periods indicated rapid oxidation (on a time scale of minutes) of $SO_2$ from this local source





which was emitted near the ground.

# 1   Introduction

Sulfate aerosols are known to impact ecosystems and climate through their deposition and radiative effects. The deposition of
sulfate aerosols can cause acidification of soils and lakes (Gerhardsson, 1994) and their direct and indirect radiative effects can
change the radiative budget at regional scales and alter climate (IPCC, 2001).

Sulfate aerosols can be primary or secondary. Primary particles are emitted directly from the surface to the atmosphere but
secondary particles are formed in the atmosphere through gas to particle conversion. The majority of anthropogenic and natural
sulfur is emitted as $SO_2$ or oxidized to $SO_2$ in the atmosphere (Berresheim et al., 1995, 2002; Seinfeld and Pandis, 1998).
Chin and Jacob (1996) and Chin et al. (2000) reported that around 50% of the globally emitted $SO_2$ is oxidized to form sulfate
and the remainder is lost by dry and wet deposition.

Dry deposition is important and gives $SO_2$ a lifetime of about 3 days for a boundary layer with 1000 m depth (Hicks, 2006;
Myles et al., 2007) and wet deposition is important for rainy days or days with fog. The lifetime of $SO_2$ is usually shorter
than 3 days because of the oxidation of $SO_2$ to sulfate. Researchers proposed a variety of lifetimes for $SO_2$ in the atmosphere
ranging from several hours to several days. As an example, in situ measurements by Hains (2007), in the eastern US found an
$SO_2$ lifetime of $19 \pm 7$ h and GEOS-chem simulations suggest a value of 13 h during summer for the same location.

The oxidation of $SO_2$ to sulfate is an important factor in determining the role of sulfate aerosols in the environment. A detailed
understanding of the different $SO_2$ oxidation pathways and their relative importance is critical for accurate estimation of the
sulfate distribution and its impact on the climate through aerosol radiative forcing. The oil sands regions are of great interest
because of the large quantities of $SO_2$ emissions. Therefore a comprehensive knowledge of $SO_2$ oxidation pathways important
in this region is useful to identify where and how atmospheric sulfur species are transported and contribute to aerosol formation,
growth and deposition.

Oil sands extraction and upgrading processes can be a source of sulfate aerosols, $SO_2$ and potential oxidants. The major
sources of $SO_2$ emissions in the Athabasca oil sands region are upgrading and energy production operations (Kindzierski and
Ranganathan, 2006).Simpson et al. (2010) measured the average background boundary layer $SO_2$ to be 102 ppt. They observed
$SO_2$ enhancement over the oil sands, with a maximum value of 39 ppb. Howell et al. (2014) showed that both $SO_2$ and sulfate
contributions from the Athabasca oil sands region are significant compared to estimates of annual forest fire emissions in
Canada. Bari and Kindzierski (2017) measured the chemical composition of particulate matter in the oil sands region with
diameter smaller than $2.5 \mu m$ ($PM_{2.5}$) and reported that the most abundant component was sulfate contributing to 9 % of the
$PM_{2.5}$ mass, on average, at Fort MacKay. Proemse et al. (2012a) reported that primary sulfate in $PM_{2.5}$ contributes less than
10 % of total sulfur emissions from two major stacks in the region, so primary sulfate from the stacks is less important than
secondary sulfate downwind.

Sulfur dioxide is converted to sulfate in homogeneous and heterogeneous reactions. The oxidation pathway is a very important





factor to determine the effects of the sulfate formed.

Homogeneous oxidation of $SO_2$ in gas phase by $OH$ radicals is as follows (Tanaka et al., 1994);

$$SO_2 + OH + M \rightarrow HOSO_2 + M \tag{R1}$$

$$HOSO_2 + O_2 \rightarrow HO_2 + SO_3 \tag{R2}$$

$$SO_3 + H_2O + M \rightarrow H_2SO_4 + M \tag{R3}$$

The product is sulfuric acid which can nucleate to form new particles or condense on the surface of preexisting particles (Benson et al., 2008; Kulmala et al., 2004) which are important to the direct and indirect radiative effect. Seventeen to 36 % of

global sulfate production can be attributed to this pathway (Chin et al., 2000; Sofen et al., 2011; Berglen et al., 2004).

Heterogeneous oxidation of $SO_2$ is considered to occur primarily in cloud droplets, although oxidation on the surface of aerosols are regionally important. Eriksen (1972) showed various steps in $SO_2$ dissolution before oxidation by major oxidants which are $H_2O_2$, $O_3$, and $O_2$ catalyzed by transition metal ions (TMIs) such as $Fe^{3+}$, $Mn^{2+}$ (TMI-catalysis) in a radical chain reaction pathway (Herrmann et al., 2000) .

$$SO_2(g) \rightleftharpoons SO_2(aq) \tag{R4}$$

$$SO_2(aq) + H_2O \rightleftharpoons HSO_3^- + H^+ \tag{R5}$$

$$HSO_3^- + H^+ \rightleftharpoons H_2SO_3 \tag{R6}$$

$$HSO_3^- \rightleftharpoons SO_3^{2-} + H^+ \tag{R7}$$

$$2HSO_3^- \rightleftharpoons H_2O + S_2O_5^{2-} \tag{R8}$$

After the dissolution oxidation occurs to convert S(IV) to S(VI).

$$S(IV) + O_3 \rightarrow S(VI) + O_2 \tag{R9}$$



$$HSO_3^- + H_2O_2 \rightleftharpoons SO_2OOH^- + H_2O \qquad\qquad SO_2OOH^- + H^+ \rightarrow H_2SO_4 \qquad\qquad \text{(R10)}$$

$$S(IV) + O_2 \rightarrow S(VI) \qquad\qquad \text{(R11)}$$

Sulfur dioxide ($SO_2$) oxidation by $O_3$ and transition metal catalysis are pH dependent and become faster as pH increases but oxidation by $H_2O_2$ within normal atmospheric pH ranges is not significantly dependent on pH. (Seinfeld and Pandis, 1998). Heterogeneous oxidation produces sulfate on the surface of preexisting aerosols or in droplets. These processes can affect aerosol sizes and their residence time. Recent studies have shown that current models significantly underestimate the TMI-catalyzed pathway (Harris et al., 2013a, b; Alexander et al., 2009). Alexander et al. (2009) showed the potential importance

of TMI catalyzed oxidation pathway attributed to anthropogenic TMIs by inclusion of this pathway into GEOS-Chem. Harris et al. (2013a) measured the sulfur isotopic composition of $SO_2$ gas upwind and downwind of clouds and used the difference to calculate the fractionation occurred for in-cloud $SO_2$ oxidation and showed that $SO_2$ oxidation catalyzed by natural transition metal ions is the dominant in-cloud oxidation pathway and is underestimated by more than an order of magnitude in current atmospheric models. To the best of our knowledge there is no study to investigate the importance of TMI-catalyzed pathway

in $SO_2$ oxidation on the surface of aerosols in polluted areas.

Until recently the oxidation of $SO_2$ with $OH$ radicals has been considered as the only important gas phase oxidation pathway. However, it has been suggested in the recent literature that oxidation of $SO_2$ by Stabilized Criegee Intermediates (sCI) can be a significant additional oxidation pathway (Berndt et al., 2012; Boy et al., 2013; Kim et al., 2015). Stabilized Criegee intermediates are thought to be formed in the atmosphere mainly through ozonolysis of unsaturated hydrocarbons particularly

alkenes (Boy et al., 2013; Welz et al., 2012). sCIs formed from the ozonolysis of alkenes are known to oxidize $SO_2$. The rate constants of the reaction of sCIs and $SO_2$ are controversial but researchers agree that the reaction is faster than what has been previously thought (e.g. $6 \times 10^{-13} \frac{cm^3}{molecule.s}$ and $8 \times 10^{-13} \frac{cm^3}{molecule.s}$ for Criegee intermediates originating from the ozonolysis of $\alpha$-pinene and limonene, respectively). Mauldin III et al. (2012), reported the oxidation of $SO_2$ by monoterepenes by measuring $OH$ and $H_2SO_4$ simultaneously during a field study in a boreal forest and confirmed the results by laboratory

and theoretical studies. Sipilä et al. (2014) also reported experimental results for the fast oxidation of $SO_2$ with sCIs formed from monoterpenes. In this study we investigate the importance of $SO_2$ oxidation by Criegee intemediates in a polluted region with high Volatile Organic Compound (VOC) emissions.

Sulfur isotope analysis is a powerful tool to investigate $SO_2$ oxidation pathways in the atmosphere. As an example, Lin et al. (2017) used high sensitivity measurements of cosmogenic $^{35}S$ in $SO_2$ and sulfate from the ambient boundary layer over coastal

California and Tibetan plateau to identify oxidation of $SO_2$ to sulfate. The conversion rate in summer ranged from 1 to 2 days suggesting that there might be oxidation pathways which are more important than previously thought.

This study uses stable sulfur isotope measurements for $SO_2$ and size segregated sulfate aerosols. For the no tracer condition data for $\delta^{34}S$ values of potential sources in the region (Proemse et al., 2012a) and isotope fractionation data (Harris et al.,





2012) were used to investigate the importance of atmospheric sulfur oxidation pathways in the Athabasca oil sands region. The $^{34}SO_2$ condition was used to examine the rate of $SO_2$ oxidation at ground level. The sulfur dioxide ($SO_2$) to sulfate conversion ratio ($F(s) = \dfrac{[SO_4]}{[SO_4]+[SO_2]}$) was also used as a tool to investigate the possible $SO_2$ oxidants in the region for a $^{34}SO_2$ tracer scenario and in the case where no tracer was present.

## 2  Sulfur isotopes

Stable sulfur isotopes can be used to investigate sulfur sources, transport and chemistry. The importance of different oxidation pathways and the effects of aerosols on the environment can be determined by these measurements. Sulfur has four stable isotopes:$^{32}S, ^{33}S, ^{34}S$, and $^{36}S$. The isotopic composition of a sulfur sample is described using the delta notation:

$$\delta^x S(‰) = (\frac{(\frac{n(^xS)}{n(^{32}S)})_{Sample}}{(\frac{n(^xS)}{n(^{32}S)})_{V-CDT}} - 1) \times 1000 \qquad (1)$$

Where n is the number of atoms,$^xS$ is the heavy isotope and V-CDT is the international sulfur isotope standard, Vienna Canyon Diablo Troilite, with the isotopic ratio of $R^{34} = \dfrac{^{34}S}{^{32}S} = 0.044163$, $R^{33} = \dfrac{^{33}S}{^{32}S} = 0.007877$ (Ding, 2001) and $R^{36} = \dfrac{^{36}S}{^{32}S} = 1.05 \times 10^{-4}$. For the purpose of this paper we only analyze $\delta^{34}S$ values and use $\delta^{33}S$ values to find enrichment of samples.

The isotopic composition ($\delta^{34}S$) of major sources of atmospheric sulfur in the Athabasca oil sands region have been measured by Proemse et al. (2012a). They reported sulfur isotope values for bitumen,$+4.3 \pm 0.3$ ‰, untreated oil sand, $+6.4 \pm 0.5$ ‰ and the isotopic composition of products such as $(NH_4)_2SO_4$ which is produced in flue gas desulfurization (FGD) ,$+7.2$ ‰ , Coke $+4.0 \pm 0.2$ ‰ and elemental sulfur, $+5.3 \pm 0.5$ ‰ . Also $\delta^{34}S$ for primary sulfate with $D < 2.5\mu m$ are reported as having values between +7.0 to +7.8 ‰ with an average of $+7.3 \pm 0.3$ ‰,and between +6.1 to +11.5 ‰ with an average of $+9.4 \pm 2‰$ for two of the largest emitters in the region. These two stacks are 12.2 km and 19.4 km south and south east of AMS13 respectively.

In addition to S emissions from oil sands processing, significant aerosols can potentially be produced from vehicle exhaust. Norman et al. (2004a, b) determined $\delta^{34}S$ values for total sulfur in gasoline and diesel fuels which are used in Alberta and British Columbia (+9 ‰), and the diesel additive molybdenum sulfide (+5 ‰), and crude oils (0 ‰), resulting in a $\delta^{34}S$ of +5 ‰ for $SO_2$ from engine exhausts (Norman et al., 2004b). Other S emissions in the region may result from anoxic conditions in the environment or the tailing ponds associated with sulfate reducing bacteria. Biogenic emissions such as $H_2S$ have negative $\delta^{34}S$ values which can be as negative as -30 ‰ (Wadleigh and Blake, 1999; Krouse and Grinenko, 1991).

Differing isotopic contributions from S sources can drive variations in aerosol sulfate $\delta^{34}S$ values. Another reason for $\delta^{34}S$ variation can be isotopic fractionation. The oxidation of $SO_2$ causes isotope fractionation between the products and reactants as long as the reaction is not complete. When the reactant is available as an infinite reservoir the fractionation factor is calculated





as

$$\alpha_{34} = \frac{R_{Products}}{R_{Reactants}} \tag{2}$$

where $R = \frac{^{34}S}{^{32}S}$. Following the definition for $\alpha$ used by Harris et al. (2012) for both kinetic and equilibrium reactions, $\alpha < 1$ means that the light isotopes react faster, so products are isotopically lighter than the reactant.

During this study, enriched $^{34}SO_2$ was emitted from a Chemical Ionization Mass Spectrometer (CIMS) close to the high volume sampler near the ground. Here we refer to these particular periods as CIMS-ON. The enrichment of $^{34}SO_2$ was sufficiently large that isotopic fractionation can be neglected during CIMS-ON periods. In contrast to CIMS-ON periods it is possible to examine S sources and oxidation pathways using $\delta^{34}S$ values for the CIMS-OFF periods. During $SO_2$ oxidation to sulfate, isotope fractionation occurs between reactants and products which is unique for each oxidation pathway. Temperature

dependent fractionation factors reported by Harris et al. (2012) for different $SO_2$ pathways are given below. Sulfur dioxide $(SO_2)$ oxidation by $OH$ radicals favors heavy isotopes and the fractionation decreases slightly by temperature(equation 3).

$$(\alpha - 1)(‰) = (10.60 \pm 0.73) - (0.004 \pm 0.015) \times T(°C) \tag{3}$$

Aqueous phase oxidation can occur by $H_2O_2$ and $O_3$, fractionation during this pathway (equation 4) also prefers heavy isotopes and decrease by temperature slightly.

$$(\alpha - 1)(‰) = (16.51 \pm 0.15) - (0.085 \pm 0.004) \times T(°C). \tag{4}$$

The fractionation during TMI-catalyzed oxidation pathway acts in the opposite direction to the other two pathways. TMI-catalysis is the only known oxidation pathway which favors lighter isotopes in the product sulfate and the fractionation strongly depends on temperature (equation 5).

$$(\alpha - 1)(‰) = (-5.039 \pm 0.044) - (0.237 \pm 0.004) \times T(°C). \tag{5}$$

## 3    Study site and methods

### 3.1    Field measurements

Sulfate aerosols and $SO_2$ were collected at a monitoring site next to the Wood Buffalo Air Monitoring Station 13 (AMS13) site just south of Fort MacKay in the Athabasca oil sands region from August 13 to September 5, 2013 as part of the Joint Canada-Alberta Implementation Plan for Oil Sands Monitoring (JOSM) project (Liggio et al., 2017, 2016; Phillips-Smith et al., 2017).

The location of AMS13 is shown in Fig.1. Temperature,and relative humidity were measured during the sampling period at a meteorological station 5 m above ground with a sampling interval of one minute. The WBEA data for wind speed and direction with 5 minutes time intervals were used and a time series for these data are shown in Fig.2.

   A high-volume sampler placed at ground level with a flow rate of $0.99 \pm 0.05 m^3/min$ was used to collect aerosols and $SO_2$. The high-volume sampler was fitted with a five stage cascade impactor to collect size-segregated aerosols on glass fiber filters





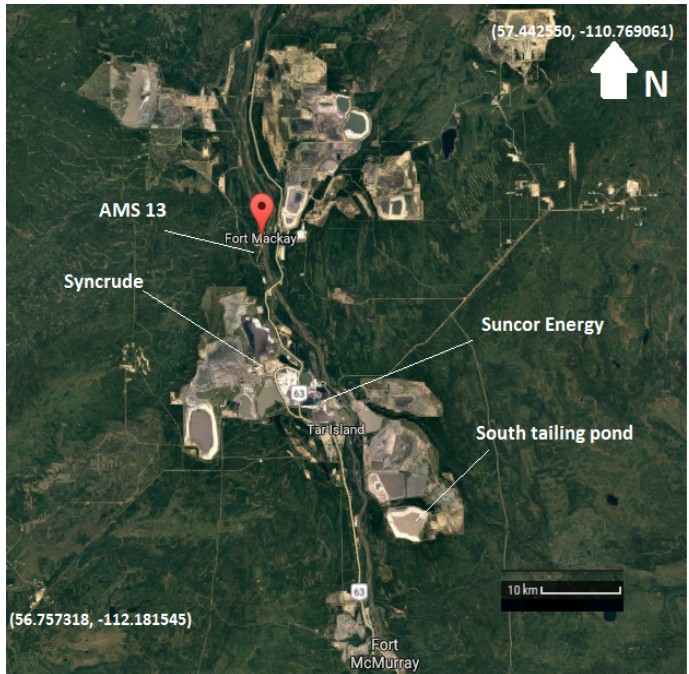

**Figure 1.** Wood Buffalo Air Monitoring Station 13 (AMS13) site, south of Fort MacKay (google map)

in five ranges of aerodynamic diameter as A (>7.2 $\mu$m), B (3.0-7.2 $\mu$m), C (1.5-3.0 $\mu$m), D (0.95-1.5 $\mu$m), E (0.49-0.95 $\mu$m). The final filter for fraction F was a $20.3 \times 25.4$ cm glass filter to collect aerosols with diameter $< 0.49\mu$m. An $SO_2$ filter pre-treated with a potassium carbonate ($K_2CO_3$) and glycerol solution was located beneath these 6 size segregated aerosol filters (Norman et al., 2004a). The sampling interval was 12 hours (daytime (5:00AM to 5:00PM) and nighttime (5:00 PM to 5:00

AM next day)) for the first twelve days except August 20 and 27 after which samples were collected for 24 hours (5:00 AM to 5:00 AM). Field blanks were collected on three separate occasions at the start, in the middle and at the end of the campaign. Filter blanks from the field were loaded and then unloaded, stored and analyzed using the same protocols as samples. The high-volume sampler was turned off during field blank sampling. Filters were stored in zip lock bags and kept in $< 4$°C and transferred to the lab for analysis.

An AF 22 Module UV florescent sulfur dioxide analyzer (AF 22 M), with an inlet 10 m away from the high-volume sampler was used to measure the atmospheric $SO_2$ mixing ratios. Since the AF 22 M was not operating for the entire period, a comparison with WBEA AMS13 $SO_2$ data was performed and based on that (r=0.98 and P-value<0.001) WBEA $SO_2$ data was used with a sampling interval of five minutes.

Ozone and $NO_2$ concentrations were measured by UV absorption and cavity ring-down spectroscopy, respectively (Paul and
Osthoff, 2010; Odame-Ankrah, 2015). Sampling intervals for $O_3$ and $NO_2$ were 10 s and 1 min and the uncertainty of measurements were $\pm 1\%$ and $\pm 10\%$ respectively. Radiometer measurements (10s) were used to determine actinic flux (J values). Iron and Mn were measured by semi-continuous X-Ray Fluorescence (XRF) measurements of metals taken every hour on a





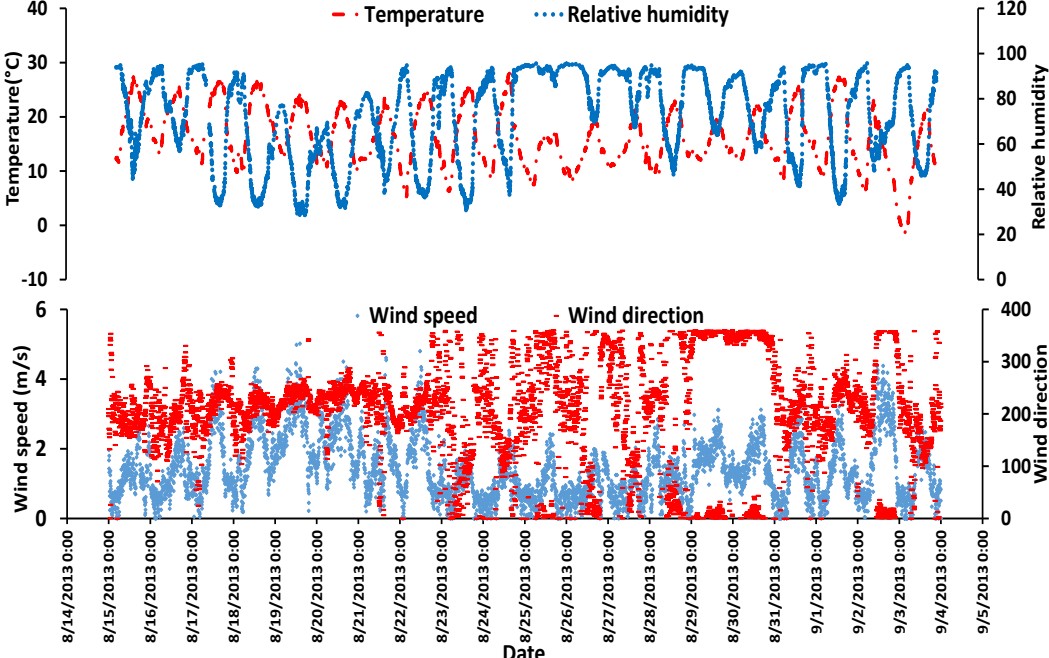

**Figure 2.** Temperature and relative humidity data with one minute sampling interval. Wind speed and wind direction with the sampling time interval of 5 minutes (data from WBEA meteorological station AMS13).

filter tape. The uncertainty of measurements is $\pm 10\%$ (Phillips-Smith et al., 2017).

Monoterpenes were measured by gas chromatography-ion trap mass spectrometer (GC-IT-MS) with variable intervals (mostly 1 hour time interval). Volatile Organic Compounds (VOCs), $C_2 - C_{12}$ were sampled in canisters over a period spanning 9:30 AM to 8:30 AM of the next day, and analyzed using gas chromatography mass spectrometery (GCMS). Detection limits for VOC measurements varied for each compound and can be found in the online JOSM database.

A Chemical Ionization Mass Spectrometer (CIMS) located 10 m away from the high volume sampler was used to measure the $OH$ concentration by Environment Canada and enriched $^{34}SO_2$ was emitted from an exhaust pipe at ground level <50 m away in an unused area containing shrubs. Enriched $^{34}SO_2$ affected a portion of our samples during CIMS-ON periods which were then used to trace the fate of local $^{34}SO_2$ emitted from the CIMS exhaust near the ground.

## 3.2 Lab measurements

Filter papers were shredded and sonicated for 30 minutes in distilled deionized water in the laboratory (200 ml for $SO_2$ and F filters and 75 ml for slotted filters). For $SO_2$ filters 1 ml of 30% hydrogen peroxide was added to $SO_2$ filter solutions to oxidize the $SO_2$ to sulfate before sonication. Filter paper fibers were removed by 0.45 mm Millipore filtration, and 10 ml of the filtrate samples was used for ion chromatography to determine the concentration of sulfate with an uncertainty of $5\%$. Remaining filtrate was treated with 0.5 ml Environmental Grade 10% $BaCl_2$, and Environmental Grade HCl was added to samples until



a pH of 3 was achieved. The samples were then heated to facilitate precipitation of $BaSO_4$. Barium sulfate was isolated by millipore filtration, and dried. Samples were packed into tin cups and analyzed with a PRISM II continuous flow isotope ratio mass spectrometer (CF-IRMS) to obtain $\delta^{34}S$ values (relative to V-CDT). The uncertainty in measuring $\delta^{34}S$ is $\pm 0.3\%$ which is determined as the standard deviation ($1\sigma$) of $\delta^{34}S$ for several standard runs. $\delta^{34}S$ measurements were blank corrected using

the sulfur concentration and $\delta^{34}S$ values for field blanks. Insufficient sulfate was present for some samples after concentration blank correction. Therefore $\delta^{34}S$ values are reported from some samples which are not isotopically blank corrected. These samples are indicated with a $*$ in tables.

The PRISM II continuous flow isotope ratio mass spectrometer measures $\delta^{34}S$ and $\delta^{33}S$ simultaneously and the values should be related according to the Mass Dependent Fractionation (MDF) relation ($\delta^{33}S \sim 0.51\delta^{34}S$). For this experiment some of the

samples are enriched in $^{34}S$ and they are identified by the use of the MDF relation between $\delta^{34}S$ and $\delta^{33}S$ of the standards for the same run. $\delta^{33}S/\delta^{34}S$ was averaged for standards for each run and $(\frac{\delta^{33}S}{\delta^{34}S}) - 2\sigma$ was used as a cutoff criteria and data falling below this criteria were tagged as enriched.

Enriched $^{34}SO_2$ for samples collected during the CIMS-ON periods in CF-IRMS measurements can be problematic due to carryover from one sample run to the next. Care was taken to analyze sufficient standards and blanks between enriched samples

(CIMS-ON periods) to ensure carryover was minimal. Little to no deviation in standards and blanks was apparent after enriched $\delta^{34}S$ values from CIMS-ON periods were analyzed. In this paper uncertainties are reported as $1\sigma$ standard deviation.

### 3.3 Natural tracer experiment

#### 3.3.1 Sulfur $^{34}S$ release

An $OH$-reactivity CIMS was located at the study site about 10 m away from the high volume sampler with an exhaust <50 m

away at ground level. The CIMS was operated between Aug 12, 12:00 PM to Aug 14, 12:00 PM and Aug 20th 12:00 PM to Sep 7th 9:45 AM. Ten sccm of $0.9\%$ $^{34}SO_2$ in $N_2$ was added to the CIMS and this flow was added to a flow of 30 SLPM of sample air which causes 3ppm mixing ratio for $^{34}SO_2$ in the sample flow. $^{34}SO_2$ reacts with $OH$ to form $H_2^{34}SO_4$ which is ionized by $NO_3^-$ to form $H^{34}SO_4^-$ and $SO_4^{2-}$ ions that are detected at $m/z = 99$ and $m/z = 49$ in the negative ion spectrum of the mass spectrometer. An excess amount of $^{34}SO_2$ compared to the required $^{34}SO_2$ to complete titration of $OH$ in the

sample flow was used for ambient air $OH$ measurements. Almost all of the flow entering was exhausted by the instrument which contains excess $^{34}SO_2$ and formed $H_2^{34}SO_4$. Calculations show that $3.7 \times 10^{-6}$ moles of $^{34}SO_2$ entered the system per minute. If the maximum $OH$ concentration in the atmosphere is considered to be $10^7 molecules/cm^3$ there will be about $4.9 \times 10^{-13}$ mole of $OH$ in the sample air in one minute. One mole of $OH$ reacts with one mole of $^{34}SO_2$ according to equations (R1), (R2) and (R3) and one mole $H_2^{34}SO_4$ is formed. So in one minute $n_{^{34}SO_2} = (7.4 \times 10^6)n_{H_2^{34}SO_4}$. Some of the

formed $H_2^{34}SO_4$ is also lost by wall loss in the instrument so the majority of the exhaust is in the form of $^{34}SO_2$. The CIMS exhaust flow was dumped into the main exhaust manifold which was directed to the east of the site (<50 m), along with all the other instrument exhausts. For the periods when the CIMS was operational (CIMS-ON) significant $^{34}S$ isotope enrichment was observed; therefore samples were divided into two sets, CIMS-ON and CIMS-OFF.





The first set is for samples collected during the shut down periods of the CIMS (CIMS-OFF) used to investigate the isotopic composition of size segregated sulfate aerosols and $SO_2$ in the region and the possible sources and formation pathways of sulfate aerosols. The second set is for samples (CIMS-ON) affected by enriched $^{34}S$ and is not used as indicators of sulfur isotopic composition of sulfate aerosols in the region. Instead the enriched $^{34}SO_2$ is used as a natural tracer to follow the fate

of $SO_2$ emitted from a local source and its oxidation near the ground.

### 3.3.2 Sulfur conversion ratio

In this paper we use the sulfur conversion ratio which is defined as the portion of $SO_2$ which is converted to particulate sulfate, and is defined as

$$F(s) = \frac{[SO_4]}{[SO_4] + [SO_2]}. \tag{6}$$

In this formula $[SO_4]$ is the concentration of sulfate aerosols with D<0.49 $\mu$m which are mostly secondary sulfate aerosols. When more oxidants are available, more $SO_2$ can be converted o particulate sulfate so F(s) increases by increasing the amount of available oxidants. Therefore significant positive correlation between F(s) and other compounds may be indicator of the importance of that compound in $SO_2$ oxidation. This formula can be used for both CIMS-ON and CIMS-OFF periods since the number of enriched molecules reaching the high volume is very small and can't change the ratio. The number of enriched

molecules reaching the high volume is calculated using equations described in section 3.3.3 and the percentage of enriched molecules in comparison to the total sulfur concentration is reported in table A1.

### 3.3.3 Concentration of $^{34}S$ enriched molecules

The concentration of enriched molecules as $^{34}SO_2$ and $^{34}SO_4$ are calculated since both isotope and concentration data are

available for $SO_2$ and sulfate samples during CIMS-ON, using the following equations. Isotope ratio (R) values show the ratio of sulfur isotopes to the most abundant isotope which is $^{32}S$ for sulfur isotopes.

$$R^{34} = n^{34}S/n^{32}S \tag{7}$$

$$R^{33} = n^{33}S/n^{32}S \tag{8}$$

$$R^{36} = n^{36}S/n^{32}S \tag{9}$$

$$R^{34}_{enriched} = (n^{34}S + n^{34}S^*)/n^{32}S \tag{10}$$





$$n^{32}S + n^{33}S + n^{34}S + n^{36}S + n^{34}S^* = S_{total} \qquad (11)$$

in which $n^{34}S^*$ is the number of $^{34}S$ atoms reaching the filter from the CIMS exhaust and $S_{total}$ is the total number of sulfur atoms on the filter. $R^{34}$ value is calculated as the average of $R^{34}$ values for samples without enrichment. There are $R^{33}$ data

available from the IRMS but the uncertainty is high and we use the value for the international standard for sulfur V-CDT. $^{36}S$ is included in calculations since the amount of $^{34}S^*$ from the CIMS exhaust is on the same order of magnitude. $R^{34}_{enriched}$ values are available for each sample. The concentration of sulfate for each sample is available from IC and the number of sulfur atoms as $SO_2$ or sulfate can be calculated. Then the number of $^{34}S$ from CIMS is calculated and divided by the volume of total sampled air and the number of $^{34}SO_2^*$ and $^{34}SO_4^*$ molecules per $cm^3$ is calculated.

High pollutant loads combined with unusual oxidant conditions create an ideal situation to study $SO_2$ oxidation pathways. Here F(s) and stable sulfur isotope ratios are used to investigate potential sulfur sources and oxidation pathways in the Athabasca oil sands region.

## 4   Results

Criegee intermediates, ozone and role of relative humidity as well as Fe and Mn in aerosol formation (F(s)) are examined

in the sections below. Insights from isotope values for CIMS-OFF (no $^{34}SO_2$ tracer) periods are considered separately from CIMS-ON.

### 4.1   The role of Criegee intermediates in $SO_2$ oxidation

Criegee intermediates are formed from ozonolysis of alkenes (consuming $O_3$) and may oxidize $SO_2$ to sulfate increasing F(s). Therefore it is expected that correlations exist between F(s) and precursors to Criegee intermediates. Significant positive cor-

relations between F(s) and the concentration of $\alpha$-pinene, $\beta$-pinene and Limonene were observed during daytime (Fig.3). No correlation was observed between F(s) and monoterpenes during nighttime.

The concentration of monoterpenes showed negative correlation with the concentration of $O_3$. There was a power law relationship between monoterpenes and $O_3$ concentration during the daytime and a linear dependency during nighttime (Fig.4). Alkene data were available only for 24 hour periods. Most of the alkenes were found to be below the detection limit. Alkenes with

concentrations higher than the detection limit except isoprene showed significant positive correlations with secondary sulfate aerosols ($D < 0.49\mu m$) and all of them except isoprene and tetrachloroethene showed significant correlations with $SO_2$ (Table A2). Tetrachloroethene was the only alkene with significant correlation with F(s) and no correlation with $SO_2$ (Fig.5).

Aromatics such as benzene, methylbenzene, ethylbenzene, m,p-Xylene, o-Xylene and propylbenzene were found to have a significant correlation with secondary sulfate and $SO_2$. 1-methyl-4-benzene(1-methylethyl), and ethenylbenzene showed sig-

nificant correlation with F(s) with r=0.58 for both and no correlation with $SO_2$ (table A3). If the rainy day (25 August) data is



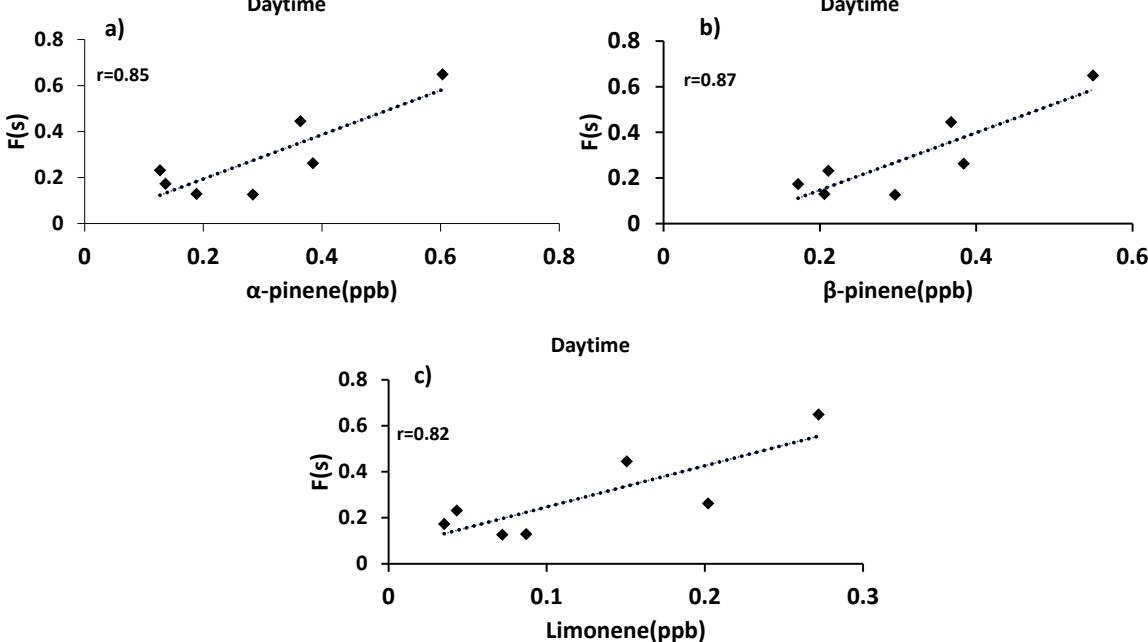

**Figure 3.** F(s) versus the concentration of a) $\alpha$-pinene, b) $\beta$-pinene and c)limonene during daytime. P-values are smaller than 0.05

eliminated the correlation is much more significant with r=0.71 and 0.66 respectively. Figure 5 shows the correlation between F(s) and tetrachloroethene, 1-methyl-4-benzene (1-methylethyl), and ethenylbenzene without the rainy day data.

## 4.2 CIMS-OFF $\delta^{34}S$ values

During CIMS-OFF periods there is no enriched $^{34}SO_2$ emission from the CIMS exhaust so $\delta^{34}S$ values reflect the sulfur
5  isotopic composition of the sulfur compounds in the region and/or fractionation as the $SO_2$ is transported from stacks to the AMS13 site. Possible oxidation pathways of $SO_2$ to sulfate is investigated using these $\delta^{34}S$ values.

### 4.2.1   $\delta^{34}S$ for $SO_2$ and size segregated sulfate aerosols

$\delta^{34}S$ values during CIMS-OFF periods for $SO_2$ and size segregated sulfate in size ranges $F_{<0.49\mu m}$, $E_{0.49-0.95\mu m}$, $D_{0.95-1.5\mu m}$, $C_{1.5-3.0\mu m}$, $B_{3.0-7.2\mu m}$ and $A_{>7.2\mu m}$ are shown in table 1.
10  Blank corrected $\delta^{34}S$ values for $SO_2$ were +5.1 and +10.8 ‰. No negative $\delta^{34}S$ values were observed for $SO_2$. If it is assumed that no fractionation occurred during formation of primary sulfate in major stacks then it is expected that $\delta^{34}S$ values for $SO_2$ would be the same as primary sulfate (with an average of; $+7.3\pm0.3$‰ and $+9.4\pm2.0$‰) in the first approximation. The $\delta^{34}S$ values of $SO_2$ are consistent with this assumption and the lowest value is consistent with a $\delta^{34}S$ value for $SO_2$ from vehicle exhaust (Table 1).



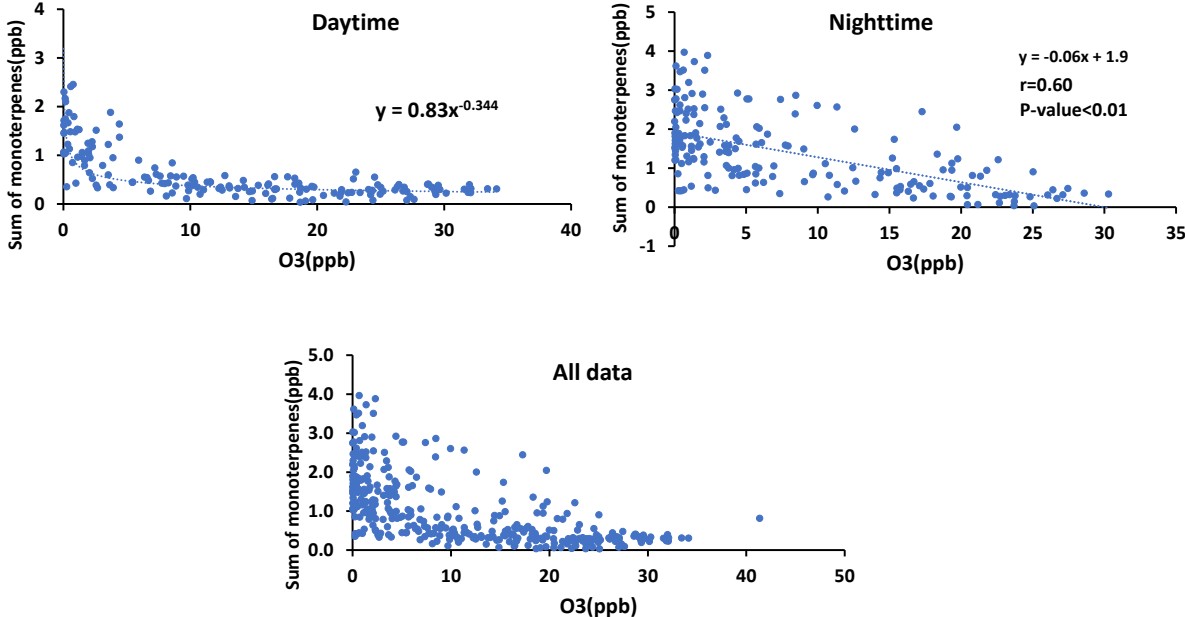

**Figure 4.** Sum of $\alpha$-pinene, $\beta$-pinene and Limonene versus ozone concentration for daytime and night time and all data

$\delta^{34}S$ values for size $F_{<0.49\mu m}$ particles, that mainly represent newly formed sulfate from $SO_2$ oxidation, ranged between +1.8 ‰ and +15.1 ‰ with an average of $+7.4 \pm 4.2$ ‰. It is worthwhile noting that although this average is the same as Proemse et al. (2012b) found for primary sulfate from the stack emissions ($+7.3 \pm 0.3$‰ and $+9.4 \pm 2$‰), there were $\delta^{34}S$ values lighter and heavier than what was expected from potential sulfur sources in the region in this size range. Therefore $\delta^{34}S$ of sulfate

5  can't be used as a quantitative indicator for industrially emitted primary sulfate as isotope fractionation may have occurred as the stack emissions ($SO_2$) were transported to the AMS13 site. Sulfate particles in this size range are predominantly secondary; therefore this data can be used to investigate the importance of different $SO_2$ oxidation pathways during transport. Particles in larger size ranges (E, D, C, B and A) may contain more primary sulfate and had lower $\delta^{34}S$ values in comparison to the $F_{<0.49\mu m}$ size range. There were no negative values for sulfate particles in the size fraction $F_{<0.49\mu m}$ that is associated

10  with $SO_2$ oxidation but negative values were observed for the size fraction $E_{0.49-0.95\mu m}$ and all the available data for blank corrected size fractions $A_{>7.2\mu m}$ and $B_{3.0-7.2\mu m}$ were negative. There was a tendency to lighter $\delta^{34}S$ values for larger sulfate particles as shown in figure 6.

   Although the concentration of sulfate was too small to perform blank correction for some samples, they displayed the same range for $\delta^{34}S$ values as those which were blank corrected. This suggests little to no bias was introduced by blank correction.




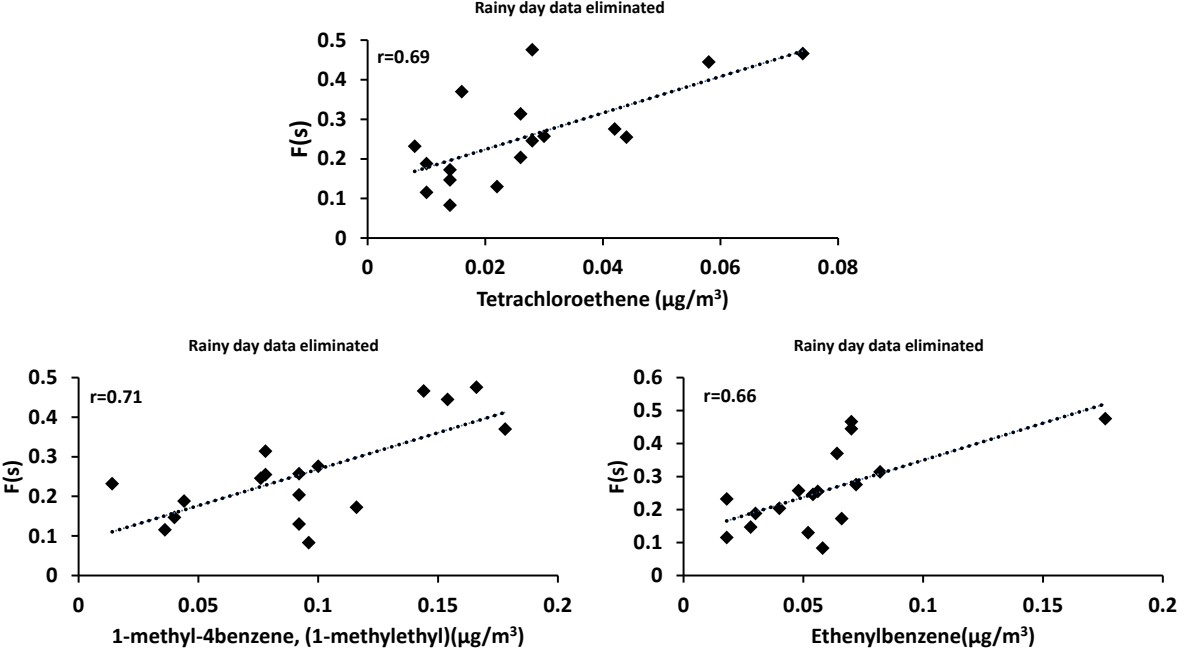

**Figure 5.** F(s) versus ethene tetrachloro and F(s) versus Benzene 1-methyl-4-(1-methylethyl), and Benzene ethenyl

**Table 1.** $\delta^{34}S$ values for $SO_2$, and sulfate aerosols in size ranges $F_{<0.49\mu m}, E_{0.49-0.95\mu m}, D_{0.95-1.5\mu m}, C_{1.5-3.0\mu m}, B_{3.0-7.2\mu m}$ and $A_{>7.2\mu m}$ during CIMS-OFF periods

| Date | $\delta^{34}S_{SO_2}(‰)$ | $\delta^{34}S_{SO_4}(F)(‰)$ | $\delta^{34}S_{SO_4}(E)$ | $\delta^{34}S_{SO_4}(D)$ | $\delta^{34}S_{SO_4}(C)$ | $\delta^{34}S_{SO_4}(B)$ | $\delta^{34}S_{SO_4}(A)$ | F(s) |
|---|---|---|---|---|---|---|---|---|
| 14Aug,pm | +10.8 | +4.6 | +2.8 | - | - | +2.1* | - | 0.77 |
| 15Aug,am | +11.1* | +6.5 | +6.5 | +0.4* | −0.38* | −0.24* | - | 0.47 |
| 15Aug,pm | - | +12.9 | +2.2* | +1.5* | - | −0.33* | -4.1 | 0.22 |
| 16Aug,am | +5.1 | +1.8 | +6.5 | +3.2* | −2.9* | −1.1* | -4.5 | 0.13 |
| 16Aug,pm | +7.4* | +8.9 | −0.2* | - | −0.4* | +3.5* | -2.1 | 0.16 |
| 17Aug,am | - | +15.1 | +6.3* | −1.1* | +1.5* | +1.8* | +0.74* | 0.13 |
| 17Aug,pm | - | +9.0* | +1.3* | +0.1* | +3.3* | +2.1* | −1.2* | 0.03 |
| 18Aug,am | +10.2* | +6.5 | -1.7 | −0.89* | −0.88* | -5.3 | −0.56* | 0.13 |
| 18Aug,pm | - | +6.1 | +2.3 | +2.1* | −1.0* | -1.8 | −2.9* | 0.10 |
| 19Aug,pm | - | +8.2 | −0.6* | +2.6* | +5.4* | +1.6* | −0.67* | 0.29 |

* not blank corrected samples





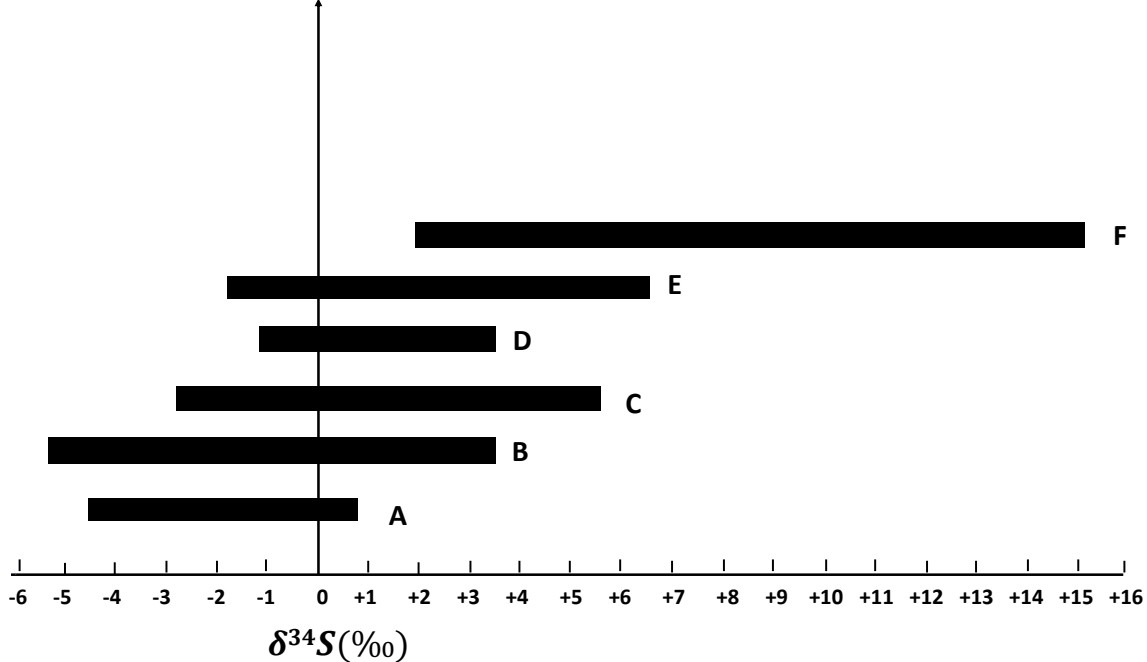

**Figure 6.** $\delta^{34}S$ ranges for $F_{<0.49\mu m}$, $E_{0.49-0.95\mu m}$, $D_{0.95-1.5\mu m}$, $C_{1.5-3.0\mu m}$, $B_{3.0-7.2\mu m}$ and $A_{>7.2\mu m}$ size ranges during CIMS-OFF periods. Not blank corrected data are also shown. As the particles become larger, $\delta^{34}S$ becomes more negative.

### 4.2.2 CIMS-OFF iron and manganese

Sulfur dioxide ($SO_2$) can be oxidized in aqueous phase by $O_2$ in presence of transition metal catalysts (TMI-catalysis) mostly $Fe^{3+}$ and $Mn^{2+}$. If this is an important oxidation pathway more secondary sulfate should be available when the concentrations of catalysts are higher. Data are available for sulfate aerosols in size fraction $F_{<0.49\mu m}$ which are mostly secondary sulfate

5   formed from the oxidation of $SO_2$ and size fraction $E_{0.49-0.95\mu m}$ which contains both secondary and primary sulfate. Positive correlations result when the concentration of sulfate in the aerosol size fractions $F_{<0.49\mu m}$ and $E_{0.49-0.95\mu m}$ are plotted against the concentration of Fe and Mn measured in $PM_2.5$ (Fig.7).

when $SO_2$ is oxidized in TMI-catalyzed pathway the sulfur isotopic composition of the formed sulfate is lighter than the isotopic composition of the reactant $SO_2$. Whenever there are more sulfate formed from this pathway lighter sulfur isotopic

10   composition is expected for sulfate. Significant anti-correlations are apparent for sulfate $\delta^{34}S$ values in size fraction $F_{<0.49\mu m}$ when plotted against Fe and Mn concentrations (Fig.8). Insufficient data were available to create similar plots for size fraction $E_{0.49-0.95\mu m}$.



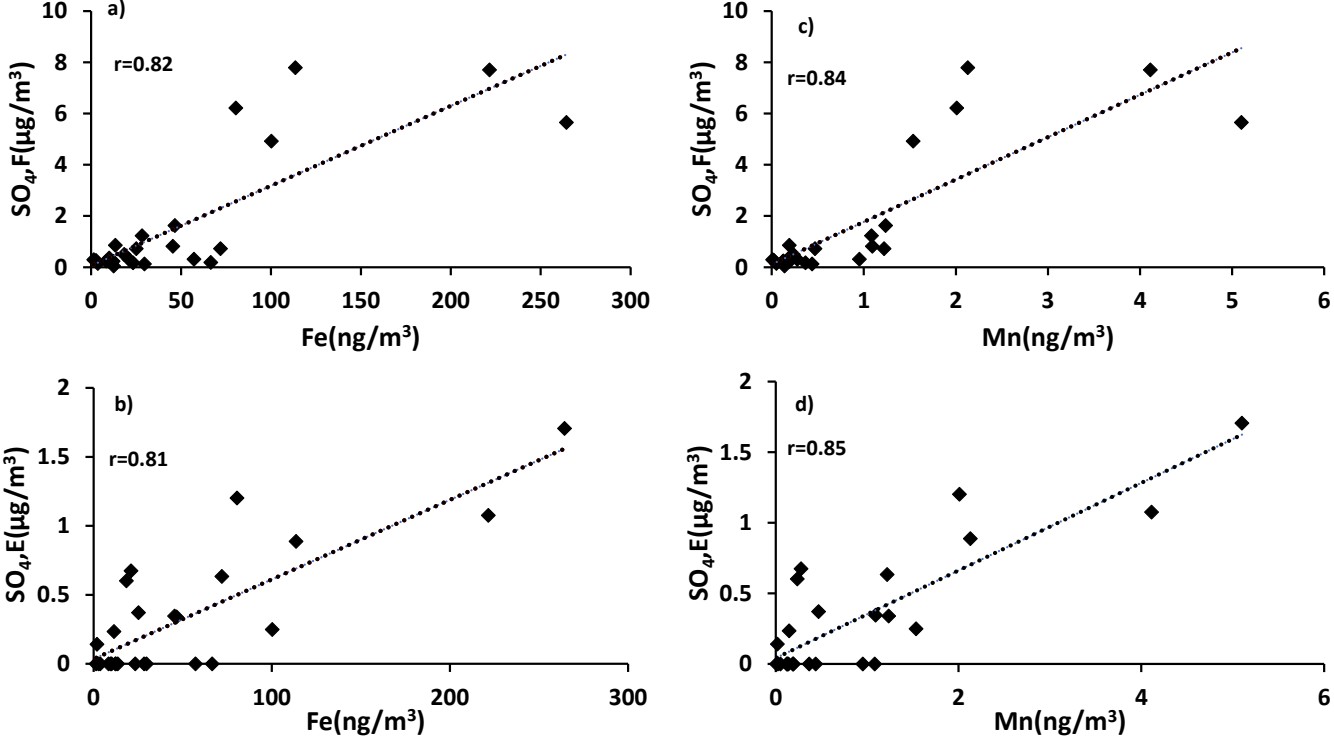

**Figure 7.** Concentration of sulfate in size range $F_{<0.49\mu m}$, and $E_{0.49-0.95\mu m}$ a), and b) versus Fe concentration and d), and e) versus Mn concentration. P-values<0.001

## 4.3 CIMS-ON $\delta^{34}S$ values

The release of $^{34}SO_2$ from the CIMS presented an ideal situation to examine $SO_2$ oxidation to sulfate under field conditions. A very unexpected result was found. $\delta^{34}S$ values for $SO_2$ samples during the periods when CIMS was operational (CIMS-ON) are shown in table 2.

5 The blank corrected data show that $\delta^{34}S$ values for enriched $SO_2$ samples were only as high as +35.6 ‰ and the values without enrichment range between +4.8 ‰ and +10.9 ‰ with an average value of $+8.3 \pm 1.8$‰.

All sulfate samples in size range $F_{<0.49\mu m}$ representing $SO_2$ oxidation during the CIMS-ON periods were blank corrected and all AM and PM samples were highly enriched in $^{34}S$; $\delta^{34}S$ values were as high as +913‰ (table 2).

Comparison between the isotopic composition of sulfate aerosols in size range $F_{<0.49\mu m}$ and $SO_2$ samples ($R_{SO_4}/R_{SO_2}$)

10 show that the sulfate particles with diameter $< 0.49\mu$m are much more enriched in $^{34}S$ from the $^{34}SO_2$ tracer released by the CIMS. The concentration of enriched sulfur as $^{34}SO_2$ and $^{34}SO_4$ molecules per $cm^3$ is also calculated as described in section 3.3.3 and the data are reported in table 2.

Sulfate aerosols during CIMS-ON periods in the size range $E_{0.49-0.95\mu m}$, $D_{0.95-1.5\mu m}$, $C_{1.5-3.0\mu m}$, $B_{3.0-7.2\mu m}$ and $A_{>7.2\mu m}$





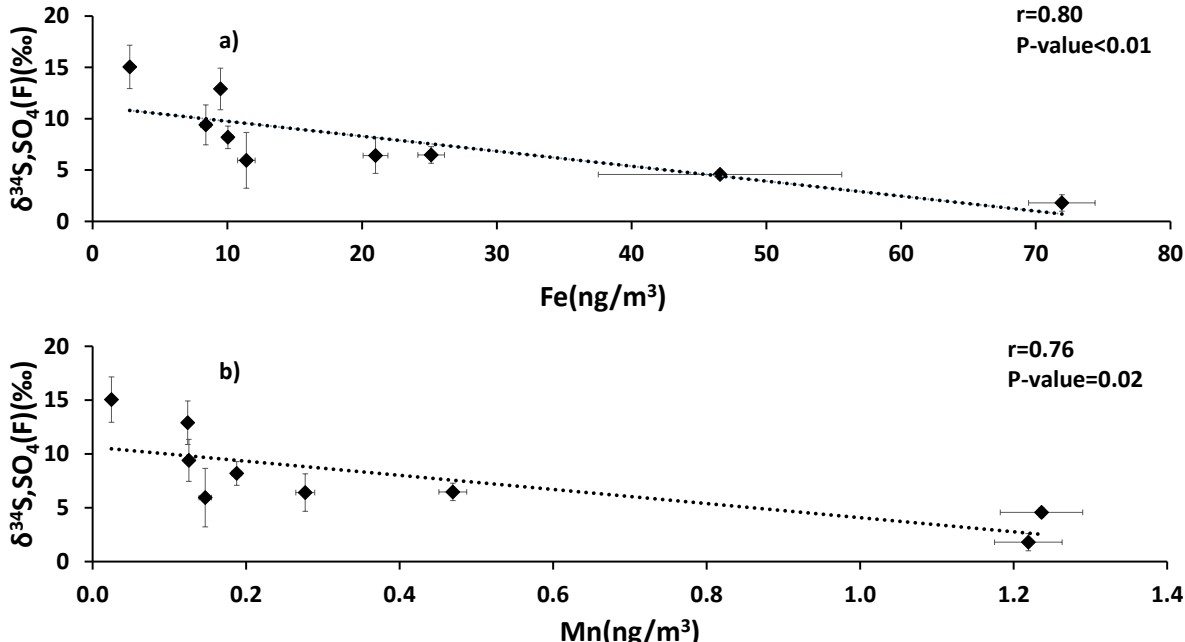

**Figure 8.** $\delta^{34}S$ values of size $F_{D<0.49\mu m}$ sulfate aerosols versus the concentration of a)Fe and b)Mn

also showed enrichment for most of the samples (table 3).

Wind coming from the south east could potentially be important for enriched sulfur to reach the high volume sampler. The percentage of wind coming from 90° to 180°was calculated for each sampling interval to show the wind coming from the CIMS exhaust. No significant correlation was observed between the wind percentage coming from the direction of CIMS exhaust and

the concentration of enriched $SO_2$ or sulfate.

### 4.4   Sulfur conversion ratio (F(s))

The sulfur conversion ratio (F(s) equation (6)) may give us valuable information about the important parameters in $SO_2$ oxidation and formation of secondary sulfate aerosols. Part of the sulfate reaching the $F_{<0.49\mu m}$ filter is from the oxidation of $SO_2$ in the atmosphere and another part comes from the enriched sulfate both from $H_2^{34}SO_4$ which condenses on the surface

of preexisting aerosols or has nucleated to form new particles, and from $^{34}SO_2$ which can be oxidized homogeneously or heterogeneously. The fraction of enriched sulfate from the CIMS exhaust to the total sulfate concentration is between 0.0003 to 0.037 and imply negligible changes to F(s) values reported in table 1 and 2. The isotope data also demonstrate a pronounced lack of $^{34}SO_2$ that supports the supposition that the majority of the enriched sulfate comes from oxidation of $^{34}SO_2$. As a result F(s) is not affected by enriched sulfate emissions and reflects conversion of $SO_2$ to sulfate. The highest F(s) value corresponds

to 25 Aug AM which was a rainy day.





**Table 2.** $\delta^{34}S$ for $SO_2$ and sulfate with diameter $< 0.49\mu$m, Ratio of sulfate to $SO_2$ isotope ratio, and the number of enriched sulfur molecules per $cm^3$ for $SO_2$ and sulfate during CIMS-ON periods. Enriched samples were selected by comparing the mass dependent fractionation relation between $\delta^{34}S$ and $\delta^{33}S$ for the sample and standards at the same run

| Date | $\delta^{34}S_{SO_2}(‰)$ | $\delta^{34}S_{SO_4}(‰)$ | $\dfrac{R_{SO_4}}{R_{SO_2}}$ | $\dfrac{n^{34}S^*(SO_2)}{V_{air}}(\dfrac{molecules(S)}{cm^3})$ | $\dfrac{n^{34}S^*(SO_4)}{V_{air}}(\dfrac{molecules(S)}{cm^3})$ | F(s) |
|---|---|---|---|---|---|---|
| 13Aug,am | $+18.6^{*,a}$ | $+155.8$ | - | - | 17770.33 | - |
| 14Aug,am | - | $+47.1$ | - | - | 40215.7 | - |
| 20Aug,daily | $+7.2^*$ | $+181.9$ | - | - | 6203.75 | 0.23 |
| 21Aug,pm | - | $+441.6$ | - | - | 10439.75 | 0.13 |
| 22Aug,am | - | $+409.5$ | - | - | 26057.18 | 0.23 |
| 22Aug,pm | $+12.2^a$ | $+572.6$ | 1.5536 | 281.11 | 16709.76 | 0.13 |
| 23Aug,am | $+4.8$ | $+27.8$ | 1.0229 | 0 | 20961.74 | 0.20 |
| 23Aug,pm | $+18.4^a$ | $+15.5$ | 0.9971 | 2867.38 | 6603.79 | 0.38 |
| 24Aug,am | $+10.9$ | $+26.8$ | 1.0157 | 0 | 14432.8 | 0.26 |
| 24Aug,pm | $+19.6^a$ | $+88.7$ | 1.0677 | 388.88 | 8693.61 | 0.43 |
| 25Aug,am | $+8.6$ | $+33.1$ | 1.0242 | 0 | 4136.85 | 0.65 |
| 25Aug,pm | $+8.4$ | $+74.4$ | 1.0655 | 0 | 2556.45 | 0.33 |
| 26Aug,am | $+7.7$ | $+21.5$ | 1.0137 | 0 | 20411.77 | 0.45 |
| 26Aug,pm | $+21.2^a$ | $+48.2$ | 1.0264 | 667.18 | 3457.02 | 0.44 |
| 27Aug,daily | $+8.4$ | $+21.5$ | 1.0129 | 0 | 11070.78 | 0.47 |
| 28Aug,am | $+13.0^*$ | - | - | - | - | 0.36 |
| 28Aug,pm | $+10.2^*$ | $+298.1$ | - | - | 11188.06 | 0.23 |
| 29Aug,daily | $+10.0$ | $+56.9$ | 1.0464 | 0 | 2010.57 | 0.25 |
| 30Aug,daily | $+7.8^*$ | $+364.9$ | - | - | 8046.98 | 0.13 |
| 31Aug,daily | $+35.6^a$ | $+312.2$ | 1.267 | 395.58 | 33841.69 | 0.48 |
| 1Sep,daily | - | $+913.3$ | - | - | 17129.41 | 0.17 |
| 2Sep,daily | $+6.9^*$ | $+735.9$ | - | - | 12055.28 | 0.15 |
| 3Sep,daily | $+7.8$ | $+24.6$ | 1.0167 | 0 | 11146.63 | 0.20 |
| 4Sep,daily | $+26.94^a$ | - | - | 13091.6 | - | - |

a tagged as enriched

* not blank corrected samples. These are only shown for comparison, no calculation has been done using these values.

F(s) (CIMS-ON and CIMS-OFF) is plotted versus relative humidity and significant positive correlation was observed for daytime and nighttime and daily samples with the same dependency. F(s) values are usually higher during the daytime so the daytime plot has a higher intercept and is above the nighttime plot (Fig.9). In the troposphere, the $OH$ radical is produced mainly from photolysis of $O_3$ to $O(^1D)$ and subsequent reaction with water vapor. If a steady state in $O(^1D)$ is assumed with





**Table 3.** $\delta^{34}S$ for sulfate in size ranges $E_{0.49-0.95\mu m}$, $D_{0.95-1.5\mu m}$, $C_{1.5-3.0\mu m}$, $B_{3.0-7.2\mu m}$, and $A_{>7.2\mu m}$ during CIMS-ON periods

| Date | $\delta^{34}S_{SO_4}(E)(‰)$ | $\delta^{34}S_{SO_4}(D)(‰)$ | $\delta^{34}S_{SO_4}(C)(‰)$ | $\delta^{34}S_{SO_4}(B)(‰)$ | $\delta^{34}S_{SO_4}(A)(‰)$ |
|---|---|---|---|---|---|
| 13Aug,am | $+25.7^*$ | $+31.5^*$ | $+45.2^*$ | $+54.1$ | $+23.1$ |
| 14Aug,am | $+28.5^*$ | $+59.1$ | $+75.1$ | $+65.5$ | $+49.0$ |
| 20Aug,daily | $+66.2$ | $+28.5^*$ | $+35.2^*$ | $+30.3^*$ | $+30.8^*$ |
| 21Aug,pm | $+31.0$ | $+34.5^*$ | $+32.8^*$ | $+33.3^*$ | $+31.5$ |
| 22Aug,am | $+76.2$ | - | $+111.4^*$ | $+110.3^*$ | $+46.1^*$ |
| 22Aug,pm | $+55.8^*$ | - | $+61.4^*$ | $+67.5$ | $+83.9^*$ |
| 23Aug,am | $+37.4$ | $+29.9$ | $+31.1$ | $+23.7$ | $+28.7$ |
| 23Aug,pm | $+15.9$ | $+12.5$ | $+11.8^*$ | $+9.3^*$ | $+6.8^b$ |
| 24Aug,am | $+8.0^b$ | - | $+29.2$ | $+18.5$ | $+10.2^b$ |
| 24Aug,pm | $+55.6$ | $+15.9^*$ | $+21.6^*$ | $+22.1^*$ | $+39.5$ |
| 25Aug,am | $+12.0^*$ | $+15.5^*$ | $+12.2^*$ | $+16.9$ | $+22.9^*$ |
| 25Aug,pm | $+27.7^*$ | $+21.3^*$ | $+18.5^*$ | $+16.9^*$ | $+32.0^*$ |
| 26Augam | $+15.8$ | $+19.7$ | $+41.4$ | $+31.9$ | $+28.9$ |
| 26Aug,pm | $+11.5^b$ | - | $+19.7^*$ | $+15.3^*$ | $+26.1^*$ |
| 27Aug,daily | $+25.6$ | $+19.9$ | $+20.9^*$ | $+31.8$ | $+32.6$ |
| 28Aug,am | - | $+217.1^*$ | $+201.3^*$ | $+212.1^*$ | - |
| 28Aug,pm | $+224.8^*$ | $+310^*$ | $+211.2^*$ | $+240^*$ | - |
| 29Aug,daily | $+80.3^*$ | $+98.5^*$ | $+85.1^*$ | $+71.9^*$ | - |
| 30Aug,daily | $+188.9^*$ | - | $+194.9^*$ | $+176.5^*$ | $+217^*$ |
| 31Aug,daily | $+166.8^*$ | - | $+537.7^*$ | $+341.8^*$ | $+339.5^*$ |
| 1Sep,daily | $+372.2^*$ | $+132.9^*$ | $+825.1^*$ | - | - |
| 2Sep,daily | - | - | $+483.4^*$ | - | $+274^*$ |
| 3Sep,daily | $+45.8$ | $+33.5^*$ | $+38.9^*$ | $+30.8^*$ | $+14.6^*$ |

b tagged as normal

* not blank corrected samples

respect to its production and loss, the (instantaneous) daytime $OH$ production rate is proportional to $j(O^1D)\times[H_2O]\times[O_3]$. A negative correlation was observed between F(s) and this (integrated) proxy during the daytime (r=0.72, P-value<0.05) (Fig.A1). However two data points with the highest RH (25 and 26 August) drive this correlation and no correlation was observed for the remainder of the samples. No correlation was observed between F(s) and $O_3$ during daytime and nighttime.





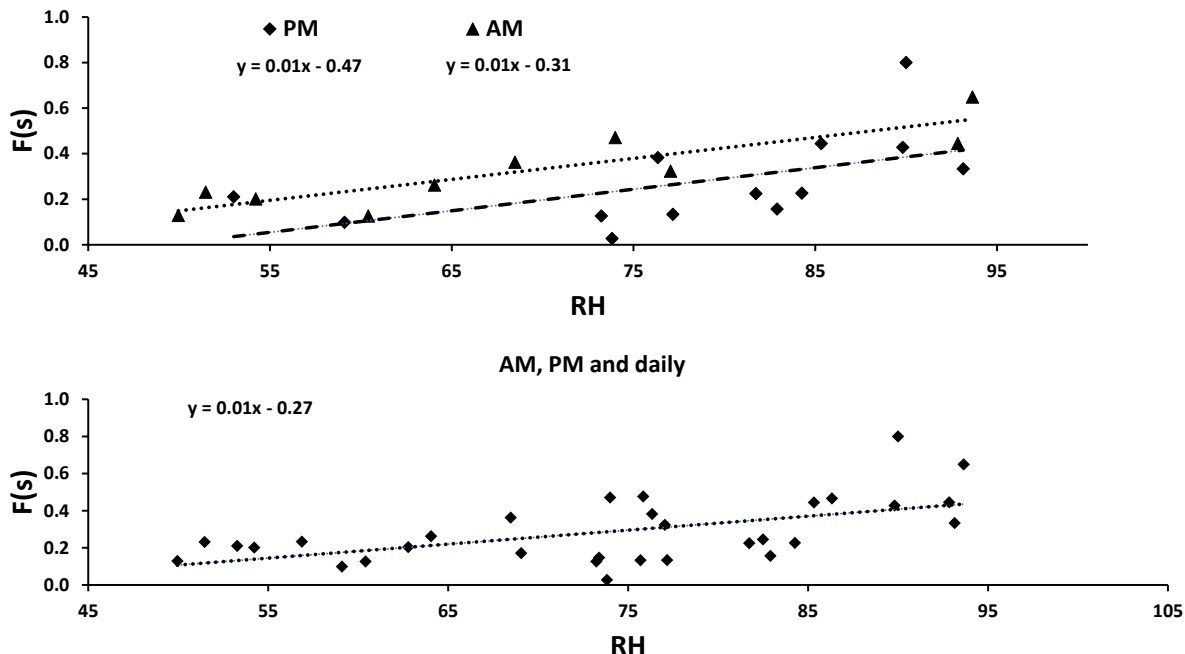

**Figure 9.** F(s) for both CIMS-ON and CIMS-OFF periods versus relative humidity. a) correlation during daytime (P-value<0.001) and night time (P-value<0.05) b) correlation for daytime and nighttime and daily data (P-value<0.001)

## 4.5   Polluted periods

Polluted periods are days when the site was impacted by plumes from major oil sands upgrading facilities (Phillips-Smith et al., 2017). Phillips-Smith et al. (2017) determined concentration time series for Positive Matrix Factorization and resolved 5 factors during the campaign. This analysis showed that August 14, 23, 24 and September 3 and 4 were days that the site was 5 impacted by the upgrader emissions. Sulfur dioxide, Fe and Mn time series are shown in figure 10. The mixing ratio of $SO_2$ is the highest on August 23 followed by September 3, August 24 and September 4. August 26 and 27 also show higher $SO_2$ mixing ratios in comparison to other days. There was a gap in $SO_2$ WBEA data during August 14 AM so the data are not complete for that day.

Available (not enriched) $\delta^{34}S$ values for $SO_2$ for these days gave an average of $+8.6 \pm 2.9 \,‰$ which was the same range as 10 for S emission from the major stacks emitting $SO_2$ in the region. If $\delta^{34}S$ values for Aug 26 and 27 are considered the average is $+8.4 \pm 2.3 \,‰$.

It is interesting to note that F(s) for daytime was higher than nighttime for all samples except polluted days, specifically August 23 and 24 (table 2). Figure 11 shows $[Fe+Mn] \times [H_2O]$ which can be used as an indicator of aqueous phase reaction of $SO_2$ with TMIS, and F(s) for nighttime samples. During nighttime on polluted days (Aug 14, 23, 24) concentrations of Fe and Mn 15 were higher than other nights and it is associated with upgrader, soil and haul road dust factors (Phillips-Smith et al., 2017). As





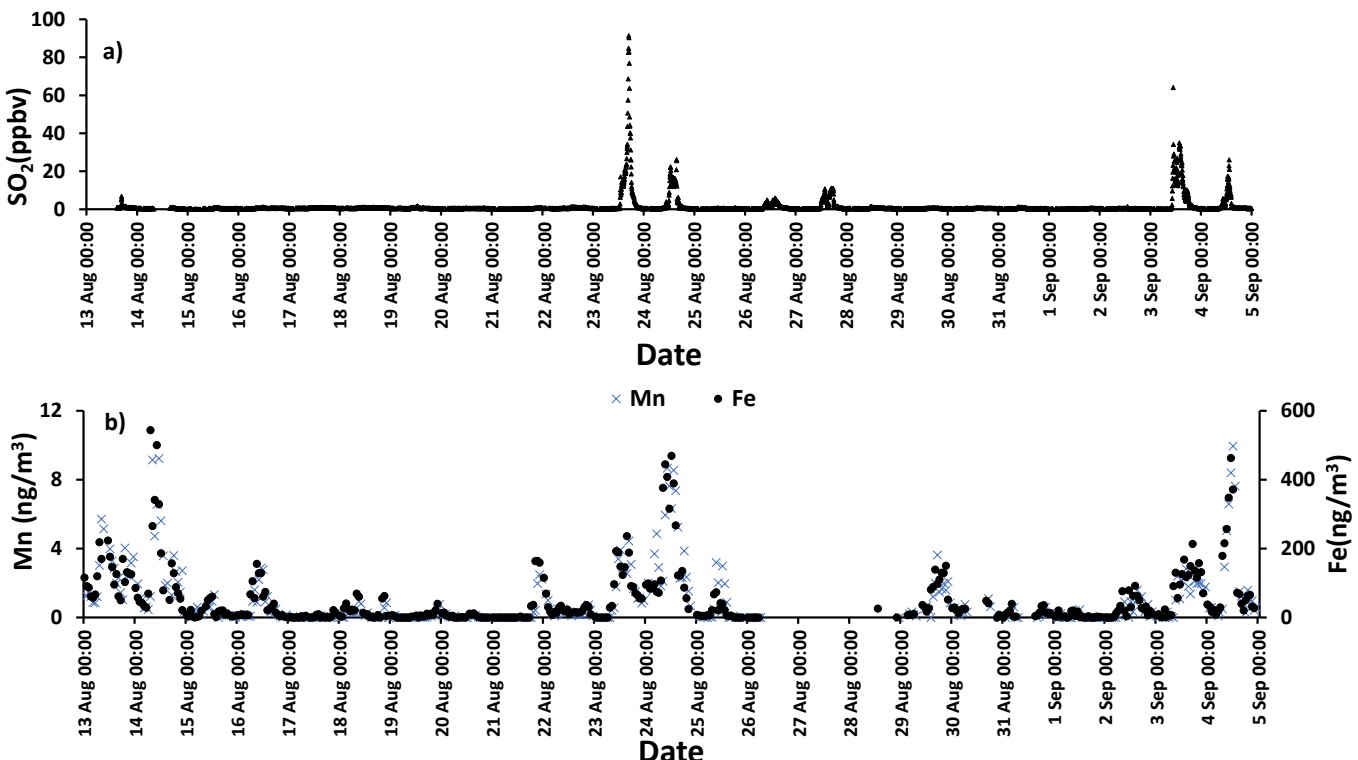

**Figure 10.** a) $SO_2$ time series. There is a gap in Aug 14 data and b)Mn(left axis)and Fe(right axis) time series.

is observed in Figure 11, F(s) was also higher during those periods. August 21 also showed high $[Fe+Mn]\times[H_2O]$ values but F(s) is not high. According to Phillips-Smith et al. (2017) during August 21 PM there was a peak for the soil factor in which Fe and Mn concentrations were high but this was only associated with a peak in the soil factor and not upgrader and haul road dust. F(s) on this night was markedly lower than on periods when upgrader and haul road dust factors were high.

5 Isotope compositions for $SO_2$ and sulfate are also interesting to consider for polluted periods. Data for both $SO_2$ and sulfate in $F_{<0.49\mu m}$ size fraction during CIMS-OFF periods were available on August 14. The $\delta^{34}S$ value for sulfate for F fraction aerosols that largely represent $SO_2$ oxidation for August 14 PM was $+4.6\,‰$ which was $6.2\,‰$ lighter than $SO_2$.

## 5 Discussion

### 5.1 $SO_2$ oxidation by Criegee intermediates

10 The proportion of $SO_4$ from $SO_2$ oxidation, F(s), during daytime, is larger than F(s) at night (Table 1 and 2) (except for polluted days). Greater vertical mixing is expected during the day than at night. Stack emissions high above ground (Gordon et al., 2017) have the potential to undergo oxidation during transport to the AMS13 site. At the same time precursors to





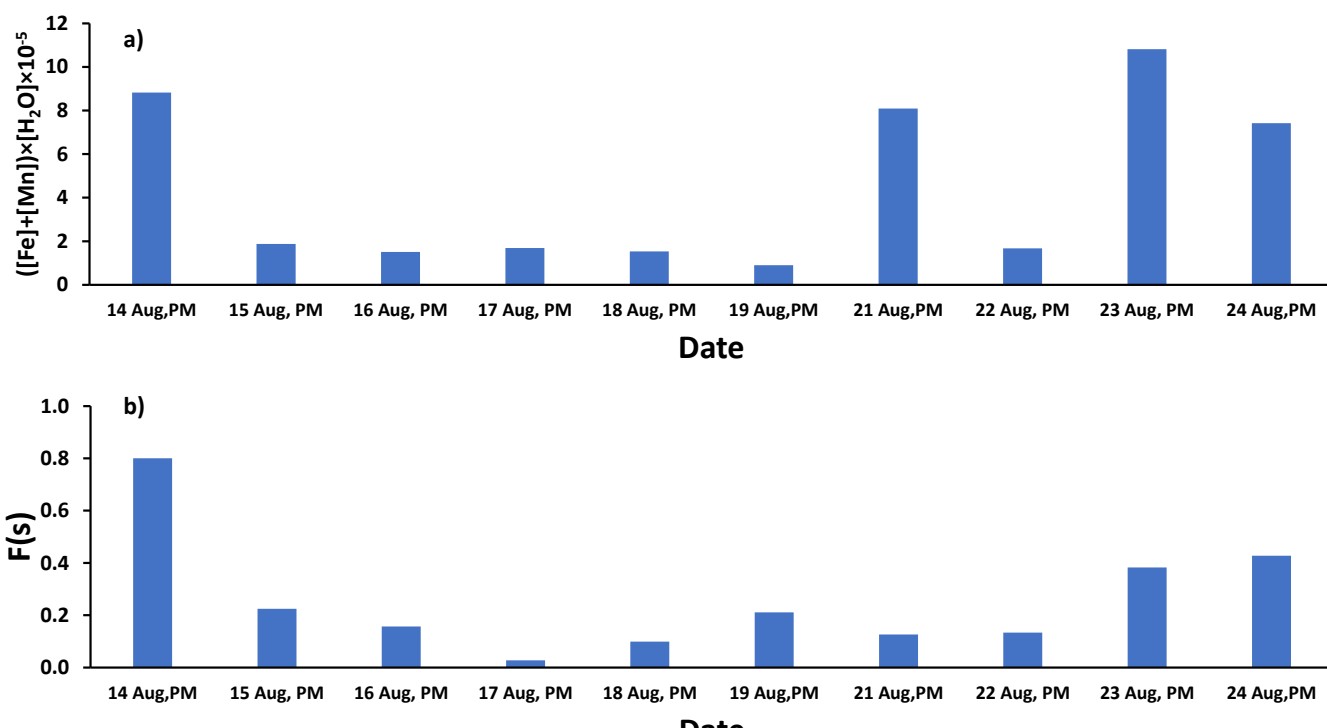

**Figure 11.** a) $([Fe] + [Mn]) \times [H_2O]$ values for nighttime samples as an indicator of the TMI-catalyzed $SO_2$ oxidation pathway and b) F(s) values for the nighttime samples

Criegee intermediates will be released and mixed upward. $H_2O$ vapor is released at ground level both at night and during the day since the region is dominated by bogs and fens. A larger F(s) during the day than at night suggests that during daytime there is an oxidation pathway in addition to aqueous phase which is the dominant pathway during nighttime. Typically it is expected that $OH$ might be the dominant oxidant during daytime. No correlation between F(s) and integrated $OH$ proxy

5  $(j(O^1D) \times [H_2O] \times [O_3])$ during days with normal RH indicates the minor importance of $OH$ oxidation pathway during daytime (Fig.A1). However a positive correlation between F(s) and $\alpha$-pinene, $\beta$-pinene and limonene during the day, but not at night (Fig.6) was observed. This combined with the loss of monoterpenes as daytime $O_3$ increased (Fig.7), suggests Criegee intermediates may be an important factor in $SO_2$ oxidation during daytime. Monoterpenes, alkenes and aromatics are oxidized by $O_3$ to form Criegee intermediates which can be stabilized and oxidize $SO_2$ to form secondary sulfate. This pathway is

10  potentially important during the day but not at night. No correlation between F(s) and monoterpenes was found at night. Many alkenes and aromatics examined may have a common source with $SO_2$ since a significant correlation ($P-value < 0.05$) was observed between them (Table A2 ,A3). The oxidation of $SO_2$ emitted from stacks in the region during transport explains the relationship between these alkenes and aromatics and secondary sulfate as the $SO_2$ can be oxidized above and below the boundary layer as the emissions mix and become entrained within the mixing layer (table A2,A3).





In contrast, specific alkenes and aromatics such as tetrachloroethene, 1-methyl-4-benzene and ethylbenzene are not likely to originate with $SO_2$ in stack emissions but their correlation with F(s) may indicate that they potentially play a role in $SO_2$ oxidation via Criegee intermediates.

## 5.2 CIMS-OFF

Aerosols in the size ranges E ,D, C, B, and A represent both primary and secondary sources of aerosols. Sulfur dioxide ($SO_2$) can be oxidized on the surface of the primary or preexisting aerosols and form secondary sulfate on their surface. Isotopically light $\delta^{34}S$ values for sulfate aerosols in these size ranges were observed during CIMS-OFF periods. These values indicate that there was no, or only a very small contribution, of primary sulfate from major stacks with average $\delta^{34}S$ values of $+7.3 \pm 0.3$ ‰ and $+9.4 \pm 2.0$ ‰ on these particles.

The sulfur isotopic composition of sulfur containing compounds in the Athabasca oil sands region were measured by Proemse et al. (2012a). All $\delta^{34}S$ values of potential sources are positive as mentioned in section 2 of this paper. The only isotopically light sulfur source in the region is $H_2S$ that could be emitted from tailing ponds.

Proemse et al. (2012b) also reported isotopically light $\delta^{34}S$ values for sulfate from bulk and through fall deposition in the Athabasca oil sands region. Summer $\delta^{34}S$ values, in particular at the industrial site W1 which is adjacent to a tailings pond and

peat pond had average $\delta^{34}S$ values of $-3.9$ ‰ and $+0.3$ ‰, respectively. This showed evidence for a contribution of sulfate from a $^{34}S$ depleted source. They suggested that the low $\delta^{34}S$ values observed for atmospheric sulfate collected at these sites was due to $H_2S$ emitted from tailing ponds in close proximity which was oxidized to $SO_2$ and subsequently formed sulfate contributed to local sulfate deposition. They also mentioned that as a result of the short lifetime of $H_2S$ (1 day (Brimblecombe et al., 1989)) $\delta^{34}S$ values of atmospheric sulfate deposition at distances >12 km from their base location do not show evidence

of $\delta^{34}S$ values <+2 ‰ for sulfate in bulk deposition and throughfall.

The AMS13 site is 12.2 km north of the base point for Proemse et al. (2012b) data and isotopically light $\delta^{34}S$ values were observed for sulfate samples. Also the sulfur isotopic composition is not expected to change significantly during oxidation of $H_2S$ to $SO_2$ (Sanusi et al., 2006; Newman et al., 1991), so if the isotopically light sulfate were formed from $H_2S$ the $\delta^{34}S$ values of $SO_2$ should have shown isotopically light $\delta^{34}S$ values. This is not observed at AMS13; the average value for $\delta^{34}S$

of $SO_2$ during CIMS-ON and CIMS-OFF (non enriched values) periods was $+7.9 \pm 2.1$ ‰. This value is in the range of $\delta^{34}S$ of primary sulfate from stack A and B (Proemse et al., 2012a). Therefore the contribution of significant amounts of $H_2S$ to isotopically light samples through an $SO_2$ oxidation pathway is ruled out and there should be another reason for $^{34}S$ values which are lighter than the expected sources in the region.

Most of the sulfate particles in $F_{<0.49\mu m}$ size range are secondary. This is supported by the reasonably strong correlation

between F(s) and RH (Fig.9). F(s) versus RH showed the same slope during daytime and nighttime which suggests that aqueous phase oxidation is the most important oxidation pathway during daytime as well as nighttime. Primary soil particles in the F size fraction were not present based on the fact that no Ca or Mg was detected in filter extracts from F filters.

Since indicators of soil dust (Ca and Mg) were not present on F samples the decrease in $\delta^{34}S$ values cannot be associated with primary sulfate from haul road dust and soil in this size fraction (Phillips-Smith et al., 2017). Instead, the lighter $\delta^{34}S$ values




for F samples in comparison to the isotopic composition of $SO_2$ for the same time interval indicates oxidation; TMI is the only fractionation pathway known for which this is feasible. Isotope fractionation will be evident when a large reservoir of $SO_2$ (e.g. from stack emission) mixes with oxidants during transport and produces accumulated sulfate product captured over 12 or 24 hours. So long as the fraction of reaction is low ($< 30\%$) the difference in $\delta^{34}S$ values for $SO_2$ and sulfate will reflect the

magnitude and direction of the fractionation process. For TMI this direction is negative and produces lighter sulfate than $SO_2$. This directly contrasts with the two other fractionating oxidation pathways for $O_3/H_2O_2$, and $OH$. Evidence that $SO_2$ released from tall stacks is transported high above the ground and mixes down toward the surface at AMS13 has been demonstrated by Gordon et al. (2017) and should provide conditions meeting the requirement for fraction of reaction $< 30\%$ described here.

Although Fe and Mn in larger size fractions where haul road dust and soil may be important, for the F samples the most likely

source is upgrader emissions which would not have Ca and Mg present. Although we have not measured Fe and Mn in the F size aerosols alone (Fe and Mn represent bulk aerosol concentrations), our results are consistent with TMI oxidation as the dominant path for $SO_2$ oxidation in the $F_{<0.49\mu m}$ size range and further research should focus on measurements for Fe and Mn in the F size fraction.

Simultaneous $\delta^{34}S$ values for $SO_2$ and sulfate in size range $F_{<0.49\mu m}$ for 14 Aug PM and 16 Aug AM showed isotopically

lighter sulfate in comparison to $SO_2$. The only oxidation pathway consistent with these results is TMI-catalysis.

The significant correlation between the concentration of sulfate in the size range $F_{<0.49\mu m}$, and $E_{0.49-0.95\mu m}$ and concentration of Fe and Mn throughout the study emphasized the importance of this oxidation pathway.

In contrast to upgrader sources for Fe and Mn on $F_{<0.49\mu m}$ aerosols, soil and haul road dust are likely present on larger size aerosols where Ca and Mg was detected. The higher concentration of Fe and Mn due to multiple sources in these larger aerosols

may increase the importance of TMI-catalyzed $SO_2$ oxidation pathway on the surface of particles. This would drive the $\delta^{34}S$ for large particles to lower values as the importance of TMI fractionation increased relative to the F particles (Fig.6).

### 5.3 CIMS-ON

No enrichment for $SO_2$ samples was observed during daytime, which indicates that no $^{34}SO_2$ reached the high volume sampler. This was not caused by the wind direction since no correlation was observed between the concentration of enriched $SO_2$

molecules (for enriched samples) and the percentage of wind coming from the location of the CIMS exhaust.

$^{34}SO_2$ could be lost through dry and wet deposition. There is only one rainy day during the campaign period (25 August) so for the other days wet deposition can't be an important loss process: instead dry deposition is the primary deposition process in the region (Proemse et al., 2012b) but the lifetime with respect to dry deposition is long (3 days (Hicks, 2006; Myles et al., 2007)). It is more likely that $^{34}SO_2$ was oxidized before reaching the high volume sampler and since the CIMS exhaust was

only 50m away from the high volume sampler and at the ground level, it may indicate that the $^{34}SO_2$ was oxidized extremely fast near ground level.

High $^{34}S$ enrichment was observed on the sulfate samples in all size ranges. Enriched sulfate can come from the oxidation of $^{34}SO_2$ and condensation of $H_2^{34}SO_4$ coming directly from the CIMS exhaust (nucleation of $H_2^{34}SO_4$ to form new particles is important for much smaller size ranges than what is talked about in this paper). During the day the concentration of $OH$ can





be between $10^6 - 10^7\ molecules/cm^3$ so a small portion of the $^{34}SO_2$ is converted to $H_2^{34}SO_4$ in the CIMS which, based on calculations, was found to be negligible. We therefore concluded that the missing $^{34}SO_2$ is oxidized before reaching the high volume sampler and is detected as $^{34}SO_4$ on size segregated aerosols.

F(s) during the day was correlated with RH and the dependency is the same as night time in which aqueous phase oxidation

is the only possible oxidation pathway of $SO_2$. This can be an indicator of the importance of aqueous phase oxidation of $SO_2$ during daytime. On hot summer days, a high concentration of $O_3$ can produce high amounts of $OH$, but no correlation was observed between F(s) and $j(O^1D) \times [H_2O] \times [O_3]$ (integrated) as a proxy of formed $OH$ for days with normal RH. This indicates the minor importance of oxidation of $SO_2$ by $OH$ radicals in the gaseous phase. There was no correlation between F(s) and $[O_3]$ and it may show that the oxidation pathway by $O_3$ as the oxidant is not very important in the aqueous phase.

These lines of evidence together point to aqueous phase oxidation of $SO_2$ during the day that is not associated with $O_3$. This suggests that aqueous phase reactions with $H_2O_2$ and $O_2$ in the presence of $Fe^{3+}$ and $Mn^{2+}$ dominate daytime $SO_2$ oxidation in the oil sands region.

A similar argument can be made for nighttime oxidation of $SO_2$ at AMS13. All blank corrected $SO_2$ samples were enriched in $^{34}S$ and all the associated sulfate samples less than $0.49\mu m$ in diameter were also highly enriched in $^{34}S$. The concentration

of $^{34}SO_4$ is consistently higher than $^{34}SO_2$. The concentration of $OH$ at night is close to zero so no $H_2^{34}SO_4$ emitted directly from the CIMS was expected: all the CIMS emission should be as $^{34}SO_2$ but the enrichment is observed on sulfate aerosols instead. This points to fast oxidation of $^{34}SO_2$ in the aqueous phase for $SO_2$ emitted close to the ground in the vicinity of the high volume sampler. The lack of correlation between F(s) and $O_3$ at night suggests $O_3$ is not important for nighttime $SO_2$ oxidation, which leaves $H_2O_2$ and $O_2$ in the presence of TMIs as the likely oxidation pathway at night as well as during the

day.

$^{34}S$ enrichment of particles for aerosols $> 0.49\mu m$ in diameter can result from the condensation of $H_2^{34}SO_4$ emitted directly from the CIMS exhaust or homogeneous and heterogeneous oxidation of $^{34}SO_2$. Very little $^{34}SO_2$ is detected and $^{34}SO_4$ is found at much higher concentrations so it is highly probable that the majority of the $^{34}S$ enrichment on particles $> 0.49\mu m$ is derived from $SO_2$ oxidation and according to the correlation of F(s) with RH, it is probable that the oxidation takes place in

the aqueous phase on the surface of pre-existing aerosols.

## 5.4   Polluted periods

During polluted periods the site is impacted by plumes from major oil sands facilities. $SO_2$ concentration is much higher during polluted days in comparison to the rest of the campaign, and the average of $\delta^{34}S$ values for $SO_2$ is $+8.6 \pm 2.9$ ‰. During these periods $SO_2$ was dominantly from major stacks based on the dominance of the upgrader factor during these periods

(Phillips-Smith et al., 2017). Consequently $\delta^{34}S = +8.6 \pm +2.9$ can be an indicator of the $\delta^{34}S$ of $SO_2$ from major stacks in the region. The concentration of Fe and Mn for measurements in $PM_{2.5}$ were also high during these periods and supports the argument presented in the last section that Fe and Mn from upgrader facilities could facilitate TMI-catalyzed oxidation on F aerosols.

The nighttime concentration of Fe and Mn were higher during polluted periods (Aug 14, 23 and 24) than other nights and




whenever $([Fe] + [Mn]) \times [H_2O]$ value was high, F(s) was also high. Again this supports aqueous phase oxidation potentially by the TMI pathway.

To summarize, isotope and concentration data for aerosols and $SO_2$ during CIMS-OFF periods and polluted days point to the oxidation of $SO_2$ through TMI-catalysis in the Athabasca oil sands region of Alberta during transport of polluted airmasses from tall stacks to the AMS13 site. Enrichment of $^{34}S$ of sulfate during CIMS-ON periods from ground-based $SO_2$ emissions points to more rapid $SO_2$ oxidation at ground level than expected.

## 6 Conclusions

Sulfur dioxide and size segregated sulfate aerosol concentrations and sulfur isotope compositions were measured during summer 2013 at the AMS13 site in the Athabasca oil sands region to investigate $SO_2$ oxidation pathways. The results show the potential importance of Criegee intermediates to daytime oxidation of $SO_2$ under highly polluted conditions. However, a considerable proportion of the $SO_2$ oxidized during both day and nighttime is likely heterogeneous, with transition metal (TMI) catalysis as the likely pathway. These inferences were made based on $\delta^{34}S$ values for $SO_2$ and secondary sulfate as well as strong relationships between the $\delta^{34}S$ values and concentration of secondary sulfate and Fe and Mn (in $PM_{2.5}$) in the smallest aerosol size fractions ($< 0.49\mu m$ and $< 0.95\mu m$). This is the first study to show that TMI-catalyzed oxidation of $SO_2$ as it is transported above and within the boundary layer is a potentially important pathway for sulfate formation in a strongly polluted environment. The fraction of secondary sulfate ($F(s) = \dfrac{[SO_4]_{<0.49\mu m}}{[SO_2] + [SO_4]_{<0.49\mu m}}$) was higher during the night than during the day for periods when the site was impacted by industrial plumes mixing downward from above. This, taken together with the high Fe and Mn concentrations at night shows the importance of aqueous phase reactions, likely by the TMI pathway as $SO_2$ is transported from the stack to the site, particularly at night. In addition, a natural tracer experiment with enriched $^{34}S$ demonstrated that oxidation of $SO_2$ at ground level was considerably faster than expected (on the order of minutes vs 10s of hours).

**Appendix A**





**Table A1.** The fraction of enriched $^{34}S$ in sulfate samples in size $F_{<0.49\mu m}$

| Date | $n^{34}S^*/totalS$ |
|------|--------------------|
| 13Aug,am | 0.006 |
| 14Aug,am | 0.002 |
| 20Aug,daily | 0.007 |
| 21Aug,pm | 0.02 |
| 22Aug,am | 0.02 |
| 22Aug,pm | 0.02 |
| 23Aug,am | 0.0008 |
| 23Aug,pm | 0.0003 |
| 24Aug,am | 0.0008 |
| 24Aug,pm | 0.003 |
| 25Aug,am | 0.001 |
| 25Aug,pm | 0.003 |
| 26Aug,am | 0.002 |
| 26Aug,pm | 0.0006 |
| 27Aug,daily | 0.0006 |
| 28Aug,am | - |
| 28Aug,pm | 0.01 |
| 29Aug,daily | 0.002 |
| 30Aug,daily | 0.015 |
| 31Aug,daily | 0.013 |
| 1Sep,daily | 0.037 |
| 2Sep,daily | 0.029 |
| 3Sep,daily | 0.0007 |
| 4Sep,daily | - |



**Table A2.** Correlation coefficients between $SO_2$, $SO_4$ and alkenes with concentration higher than the detection limit

|  | $SO_4$ | $SO_2$ | F(s) |
|---|---|---|---|
| Ethene | 0.59 | 0.52 | 0.45 |
| 1-Propene | 0.70 | 0.69 | 0.29 |
| 1-Propene,2-methyl | 0.77 | 0.68 | 0.33 |
| 1-Butene,3-methyl | 0.78 | 0.71 | 0.40 |
| isoprene | 0.02 | 0.03 | 0.002 |
| ethenetetrachloro | 0.72 | 0.40 | 0.53 |




**Table A3.** Correlation coefficients (r) between $SO_2$ , $SO_4$ and aromatics

|  | $SO_4$ | $SO_2$ | F(s) |
|---|---|---|---|
| benzene | 0.73 | 0.47 | 0.45 |
| methylbenzene | 0.76 | 0.56 | 0.44 |
| ethylbenzene | 0.85 | 0.63 | 0.42 |
| m,p-Xylene | 0.85 | 0.65 | 0.39 |
| o-Xylene | 0.86 | 0.65 | 0.41 |
| Benzene propyl | 0.80 | 0.51 | 0.48 |
| 1-methyl-4-benzene | 0.23 | 0.02 | 0.58 |
| ethenylbenzene | 0.06 | 0.07 | 0.58 |

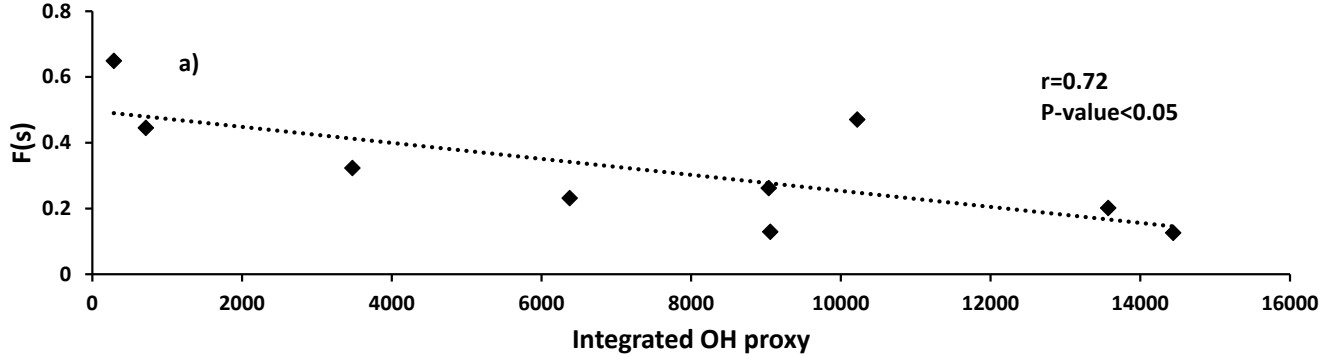

**Figure A1.** a)Sulfur conversion ratio (F(s)) versus the integrated OH proxy b)F(s) versus the integrated OH proxy when two data points for the highest RH (25 and 26 August) are eliminated



*Acknowledgements.* This project was funded by Environment Canada under Joint Canada-Alberta Implementation Plan for Oil Sands Monitoring (JOSM) and NSERC. We would like to thank Jeff Brook and Daniel Wang from Environment Canda for VOC measurements and Greg Evans and Cheol-Heon Jeong from University of Toronto for their data on Fe and Mn concentrations. We also thank Jeremy Wentzell from Environment Canada for his assistance in defining working periods of CIMS.





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
