# Peer review of "Stable sulfur isotope measurements to trace the fate of $SO_2$ in the Athabasca oil sands region"

_Atmospheric Chemistry and Physics, 2017_

## Short Comment (SC1) · 20 Jan 2018

A preliminary version of the manuscript was posted for discussion prematurely. We encourage the reviewers to review the corrected version instead. The revisions are in process and a corrected manuscript will be uploaded shortly.

---

## Referee Comment (RC1) · Anonymous Referee #2 · 2 Feb 2018

The paper presents interesting information about sulfur isotopes fractionation in the aerosols taken from a monitoring site next to the Wood Buffalo Air Monitoring Station 13 (AMS13), south of Fort MacKay in the Athabasca oil sands region from August 13 to September 5, 2013 as part of the Joint Canada. Although the analyses and data obtained are very good, they represent a short period of time and do not reflect the variability on a seasonal scale. The discussion of sulfur compounds and isotopes transformation processes does connect much to effects of natural variability and emission from land surfaces on regional scale. More information about these and impact on weather/climate and the environment are needed to motivate why this study has been done. Also, the conclusions do not show what are the new findings by the work that differ from earlier investigations. I think the paper needs to consider the points

mentioned above for possible publication in ACP.

---

## Referee Comment (RC2) · Anonymous Referee #3 · 6 Feb 2018

I have checked the response of the authors, and I think they are fine, and they answer my question and commments, and I think the manuscript could be accepted for publication.

---

## Short Comment (SC2) · 21 Feb 2018

Thanks for your comments. We have considered your comments in the corrected manuscript (supplement) as follows:

1. Although the data represent a short period of time and do not reflect the variability on a seasonal scale, Soares et al. (2018) showed that short-term measurements are more suitable for source identification. He mentioned that the source signals of NO2 and SO2 emissions are available in hourly to daily time scales and long-term observation may cause a loss in short-term variation.

2. Sulfate aerosols are known to impact ecosystems and climate through their deposition and radiative effects. The deposition of sulfate aerosols can cause acidification of

soils and lakes (Gerhardsson, 1994). Furthermore, their direct and indirect radiative effects can change the radiative budget at regional scales and alter climate (IPCC, 2001). Sulfur dioxide (SO2) is converted to sulfate in homogeneous and heterogeneous reactions. The oxidation pathway is a very important factor to determine the effects of the sulfate formed on the environment.

3.Field studies suggested that TMI-catalyzed oxidation is the dominant sulfate formation pathway in polluted environments in winter (Jacob et al., 1984, 1989; Jacob and Hoffmann, 1983). Oxygen isotope measurements of sulfate aerosols collected at Alert, Canada (82.5°N, 62.3°W) showed that TMI-catalyzed SO2 oxidation is significant during winter (McCabe et al., 2006). Recent studies have shown that the TMI-catalyzed oxidation pathway is underestimated (more than an order of magnitude) in all current atmospheric chemistry models (Harris et al., 2013a, b). For example, Harris et al. (2013a) measured the sulfur isotopic composition of SO2 upwind and downwind of clouds and used the difference to calculate the fractionation that occurred for in-cloud SO2 oxidation. They showed that SO2 oxidation catalyzed by natural TMIs on mineral dust is the dominant in-cloud oxidation pathway and is underestimated by more than an order of magnitude in current atmospheric models. To the best of our knowledge, there is no study to investigate the importance of the TMI-catalyzed pathway in SO2 oxidation on the surface of aerosols in highly polluted areas such as the Alberta oil sands region during summer.

Please also note the supplement to this comment:
https://www.atmos-chem-phys-discuss.net/acp-2017-1023/acp-2017-1023-SC2-supplement.pdf

---

## Author Comment (AC2) · 21 Feb 2018

We encourage the reviewers to review the corrected version of the manuscript uploaded as a supplement.

Please also note the supplement to this comment:
https://www.atmos-chem-phys-discuss.net/acp-2017-1023/acp-2017-1023-AC2-supplement.pdf

---

## Referee Comment (RC3) · Anonymous Referee #2 · 22 Feb 2018

Although the paper still has limited temporal extent (lacking measurements during different seasons), the corrected manuscript and the data worth publication.

---

## Author Response (AR1)

Neda Amiri
University of Calgary
2500 University Drive NW
Calgary, AB, T2N 1N4
CANADA
April 5, 2018

Attn: Editor ACS JOSM Special Issue
Re: Manuscript No. acp-2017-1023

Dear Shao-Meng Li,

The co-authors and myself thank the reviewers for their valuable insights and recommendations to make the manuscript better. The manuscript has been significantly revised and the argumentation reorganized and strengthened based on their input.

The abstract of the manuscript has been reorganized in response to reviewer 1 to make it easier to follow. More explanations about the strategy are added to the manuscript according to a comment by reviewer 1 and the organization of the paper has been changed to make it more logical. In the supplementary section in response to reviewer 1 suggestion, Figure A2 has been added to show the negative correlation between F(s) and OH production rate. The data were re-evaluated, and the correlation was driven by two points with the highest RH and were not considered to be representative. Other figures are also added to supplementary containing, F(s) vs ozone mixing ratio, and concentrations of Fe and Mn in $PM_{2.5}$ during nighttime. Sulfate concentration in size ranges F and E vs the concentration of Fe and Mn are represented in supplementary in new version and sulfate vs the addition of Fe and Mn is also added to plots. Fe and Mn vs $SO_2$, F(s) vs the concentration of two aromatics (Styrene and p-cymene), and Styrene concentration vs p-Cymene concentration plots are also added to supplementary.

In response to reviewer 2, a reference and some explanations have been added to the paper to address the concern about not reflecting the variability on a seasonal scale. The introduction is also expanded to include two paragraphs clarifying the impact on weather/climate and the environment as reviewer 2 suggested. Addressing another concern about the new findings, we have added some paragraphs to the introduction and conclusion sections of the manuscript. The remainder of the reviewer's comments have been responded to in point form as shown below.

**Report #1:**

Thank you for your valuable comments. Many of the recommended changes you mentioned in your report were addressed as described below.

Abstract:

"Sulfur dioxide oxidation by Criegee radicals may be a potential oxidation pathway during daytime. 34S enriched sulfur was emitted from a nearby Chemical Ionization Mass Spectrometer (CIMS). "

These statements do not seem connected to a plan. I would expect something like:

"To investigate the importance of Sulfur dioxide oxidation by Criegee radicals, 34S-enriched sulfur was …."

The same holds for: "Analysis of stable isotope enhancements indicated rapid oxidation of SO2 (on a time scale of minutes).", and the reason following. Please make the flow of the abstract more logical. At the end of the abstract it is not clear what is the role of Criegee radicals.

The flow of the abstract was made more logical and sentences were added to clarify the role of Criegee intermediates in $SO_2$ oxidation.

Abstract. Concentrations and $\delta^{34}S$ values for $SO_2$ and size segregated sulfate aerosols were determined for Air Monitoring Station 13 (AMS13) at Fort MacKay in the Athabasca oil sands region, northeastern Alberta, Canada as part of the Joint Canada-Alberta Implementation Plan for Oil Sands Monitoring (JOSM) campaign from Aug 13 to Sep 5, 2013. Sulfate aerosols and $SO_2$ were collected on filters using a high-volume sampler, with 12 or 24 hour time intervals. Sulfur dioxide ($SO_2$) enriched in $^{34}S$ was exhausted by a Chemical Ionization Mass Spectrometer (CIMS) operated at the measurement site and affected isotope samples for a portion of the sampling period. It was realized that this could be a useful tracer and samples collected were divided into two sets. The first set includes periods when the CIMS was not running (CIMS-OFF) and no $^{34}SO_2$ was emitted. The second set is for periods when the CIMS was running (CIMS-ON) and $^{34}SO_2$ was expected to affect $SO_2$ and sulfate high-volume filter samples. $\delta^{34}S$ values for sulfate aerosols with D>0.49 µm during CIMS-OFF periods (no tracer $^{34}SO2$ present) indicate the sulfur isotope characteristics of secondary sulfate in the region. Such aerosols had $\delta^{34}S$ values that were isotopically lighter (down to -4.5 ‰) than what was expected according to potential sulfur sources in the Athabasca oil sands region (+3.9 ‰ to +11.5‰). Lighter $\delta^{34}S$ values for larger aerosol size fractions is contrary to expectations for a primary unrefined sulfur from untreated oil sands (+6.4 ‰) mixed with secondary sulfate from $SO_2$ oxidation and accompanied by isotope fractionation in gas phase reactions with OH, or the aqueous phase by $H_2O_2$ or $O_3$. Significant anti-correlations between $\delta^{34}S$ values of dominantly secondary sulfate aerosols with D<0.49 µm and the concentrations of Fe and Mn (r = -0.80 and r = -0.76, respectively) were observed. These results indicate that $SO_2$ was oxidized by a transition metal ion (TMI) catalyzed pathway involving $O_2$ and $Fe^{3+}$ and/or $Mn^{2+}$, an oxidation pathway known to favor lighter sulfur isotopes. Analysis of $^{34}S$ enhancements of sulfate and $SO_2$ during CIMS-ON periods indicated rapid oxidation of $SO_2$ from this local source at ground level on the surface of aerosols before reaching the high-volume sampler or on the collected aerosols on the filters in the high-volume sampler. Correlations between $SO_2$ to sulfate conversion ratio (F(s)) and the concentrations of α-pinene (r = 0.85), β-pinene (r = 0.87), and limonene (r = 0.82) during daytime indicate that $SO_2$ oxidation by Criegee biradicals may be a potential oxidation pathway in highly polluted regions.

Pptv, ppbv: I understood that these units are obsolete. Ppb and ppt are preferred.
We changed pptv and ppbv to ppt and ppb.

page 2: line 16: componet component
This sentence is removed in the new version.

 page 2: line 22: OH is non-cursive, while other compounds are cursive
Change has been made. All the compounds are non-cursive in the new version.

 page 3: line 17: If I am right these processes are pH dependent. Good to mention?
Yes. pH dependence is mentioned in page 4, line 3 as follows:
The oxidation of $SO_2$ by $O_3$ and $O_2$ catalyzed by TMIs is pH dependent and becomes faster as pH increases, whereas oxidation by $H_2O_2$ within normal atmospheric pH ranges (2-7) does not depend on pH (Seinfeld and Pandis, 1998).

Page 4, line 22: if you mention the Triolite 34S/32S ratio (0.044163 (Ding, 2001), also 33 and 36 should be mentioned. Now these are mentioned on page 10 (for sulfur V-CDT with R33 = 0.007877 and R36 = 1.05 × 10−4). In the following lines, I think only 34S is mentioned. So, maybe add that you will only analyse 34S. E.g. on line 24 it is currently not clear that you talk about 34S.
33 and 36 sulfur isotopic ratio for the V-CDT standard are added to page 6, line 5 as follows;
where n is the number of atoms, $^xS$ is the heavy isotope and V-CDT is the international sulfur isotope standard, Vienna Canyon Diablo Troilite, with the isotopic ratio of $R^{34} = {^{34}S} /{^{32}S} = 0.044163$, $R^{33} = {^{33}S}/ {^{32}S} = 0.007877$ (Ding et al., 2001) and $R^{36} = {^{36}S}/ {^{32}S} = 1.05× 10^{-4}$.
We also added the following sentence to page 6, line 6 to clarify that we analyze $^{34}S$: For this paper we only analyze $\delta^{34}S$ values and use $\delta^{33}S$ values to find enrichment of samples.
Page 6, line 8 we added $\delta^{34}S$ in parenthesis to clarify this.

Page 5, line 8: When the reactant is available as an infinite reservoir, as is the case for atmospheric oxidation of SO2… When the lifetime gets very short, as suggested in this manuscript, this will certainly not be true.
The fast oxidation of $SO_2$ is only observed for a local $SO_2$ source which is at the ground level. According to F(s) data that we have for the $SO_2$ transported from the stacks to the site, $SO_2$ oxidation is not as fast as what we observe for this source at the ground level. Explanations were added to clarify that fast oxidation was observed from a local source at ground level only.
Page 1, line 19: Analysis of $^{34}S$ enhancements of sulfate and $SO_2$ during CIMS-ON periods indicated rapid oxidation of $SO_2$ from this local source at ground level on the surface of aerosols before reaching the high-volume sampler or on the collected aerosols on the filters in the high-volume sampler.
Our data do not suggest rapid $SO_2$ oxidation during plume transport and before it is entrained downward into the mixed layer, only after it has encountered conditions close to ground level in the mixed layer.

Page 10: "High pollutant loads combined with unusual oxidant conditions create an ideal situation to study SO2 oxidation pathways. Here F(s) and stable sulfur isotope ratios are used to investigate potential sulfur sources and oxidation pathways in the Athabasca oil sands region. " For me, the research strategy is still unclear after this section. The above sentence creates some clarity, but it would be good to describe in some more detail the strategy (CIMS-ON/SIMS-OFF). What is the role of the SIMS-ON measurements? How will it be used? Maybe do this also before details are given. Now the above sentence comes as an afterburner, and honestly speaking, I was quite lost even after reading the section twice.

More explanation about the strategy was added to some parts of the manuscript which are as follows:

Page 4, line 33: In this study, stable sulfur isotope values for $SO_2$ and size segregated sulfate aerosols were measured. $\delta^{34}S$ values of potential sources in the region (Proemse et al., 2012a) and isotope fractionation data (Harris et al., 2012) were used to investigate the importance of atmospheric sulfur oxidation pathways in the Athabasca oil sands region. Sulfur dioxide ($SO_2$) to sulfate conversion ratio (F(s) =$[SO_4]/([SO_4]+[SO_2])$) was also used as a tool to investigate the possible $SO_2$ oxidants in the region.

Page 6, line 30: During this study, minute quantities of $^{34}SO_2$ was emitted from a Chemical Ionization Mass Spectrometer (CIMS) exhaust 50 m away from the high-volume sampler near the ground for special periods. Here we refer to these particular periods as CIMS-ON. The enrichment of $^{34}SO_2$ was sufficiently large that isotopic fractionation can be neglected during CIMS-ON periods. However, sulfur sources and oxidation pathways can be examined using $\delta^{34}S$ values for the periods when CIMS was not operational (CIMS-OFF).

Page 9, line 17: For the periods when the CIMS was operational (CIMS-ON) significant $^{34}S$ isotope enrichment was observed; therefore, samples were divided into two sets, CIMS-ON and CIMS-OFF. The first set is for samples collected during the shutdown periods of the CIMS (CIMS-OFF). These CIMS-OFF periods were used to investigate the isotopic composition of size segregated sulfate aerosols and $SO_2$ in the region and the possible sources and formation pathways of sulfate aerosols. The second set is for samples (CIMS-ON) affected by enriched $^{34}S$ and is not used as indicators of sulfur isotopic composition of sulfate aerosols in the region. Instead the enriched $^{34}SO_2$ is used as a natural tracer to follow the fate of $SO_2$ emitted from a local ground-based source and its oxidation.

Page 9, line 30:  Since F(s) is a measure of $SO_2$ to sulfate conversion it is a measure of oxidant loading. Therefore, significant positive correlation between F(s) and other compounds may be an indicator of the importance of that compound as a tracer for $SO_2$ oxidation.

We also changed the organization of the paper in a way that is easier to follow. The changes are as follows:

Study site section is moved to section 2 before the sulfur isotopes section. Methods is section 4 and field measurements, analysis of high-volume filter samples, and natural tracer experiment

are subsections. Results is section 5. In the new version we started the result section with Sulfur conversion ratio (F(s)) and continued with $\delta^{34}S$ values for size segregated sulfate aerosols and $SO_2$ during CIMS-OFF periods. Next section in results is $\delta^{34}S$ values of $SO_2$ and size segregated sulfate aerosols during CIMS-ON periods and the result section end with the role of Criegee biradicals in $SO_2$ oxidation. Discussion is also reorganized and there are three subsections as Potential TMI-catalyzed $SO_2$ oxidation, CIMS-ON and Potential oxidation of $SO_2$ by Criegee biradicals, respectively.

Page 10, line 16: positive correlations between F(s) and the concentration of α-pinene, β-pinene and Limonene was observed during …was should be were.

We have corrected the manuscript as requested.

Page 20, line 7: Significant positive correlations between F(s) and the concentration of α -pinene (r = 0.85), β-pinene (r = 0.87) and limonene (r = 0.82) were observed during daytime (Figure 8).

Section 4.1 on page 10 is about the correlation between various kinds of double bond species and F(s). Here the idea is apparently that Criegee intermediates may oxidize SO2 to sulfate, increasing F(s). Maybe it is good to reiterate this at the start of the paragraph…

The following sentences have been added to the text.

Page 9, line 30: Since F(s) is a measure of $SO_2$ to sulfate conversion it is a measure of oxidant loading. Therefore, significant positive correlation between F(s) and other compounds may be an indicator of the importance of that compound as a tracer for $SO_2$ oxidation.

Page 20, line 6: Criegee biradicals are formed from ozonolysis of alkenes and may oxidize $SO_2$ to sulfate increasing F(s).

Page 11, line 5-6: supposed assumed

The suggested change has been made.

Page 12, line 3: If a steady state in O(1D) is assumed with respect to its production and loss, the (instantaneous) daytime OH production rate is proportional to j(O1D)×[H2O]×[O3].

Page 13, line 4: incorporated was introduced

This section of the text has been revised.

Page 8, line 30: Although the concentration of sulfate was too small to perform blank correction for some samples, they displayed the same range for $\delta^{34}S$ values as those which were blank corrected. This suggests little to no bias was introduced by blank correction.

Page 13, line 7: sulfate aerosols for size fractions sulfate in the aerosol size fractions

Page 16, line 4: Concentrations of Fe and Mn were measured in $PM_{2.5}$ particles, therefore, the concentration of sulfate in size fractions F<0.49 μm, E (0.49<D<0.95 μm), D (0.95<D<1.5 μm), and C (1.5<D<3.0 μm)  were added to find the concentration of sulfate in particles with D<3 μm.

Page 13, line 8: without mentioning the correlations with the coarser fractions, these two correlations are meaningless.

We have used different correlations as follows:

Correlation between sulfate in $PM_{2.5}$ and the concentration of Fe and Mn in $PM_{2.5}$ particles. This correlation shows that whenever the concentration of Fe and Mn in $PM_{2.5}$ is high the concentration of sulfate is also high. This can be because of the TMI catalyzed pathway of $SO_2$ to form sulfate or this can show that sulfate or $SO_2$ have a common source with Fe and Mn.

We also observed a positive correlation between sulfate in size ranges F and E with Fe and Mn. Sulfate in these size range is mostly secondary, so this can be an indicator of the importance of TMI catalyzed pathway or common source of $SO_2$ and Fe and Mn.

Finally, a negative correlation was observed between $\delta^{34}S$ values for size fraction F and the concentration of Fe and Mn which shows that when the concentration of TMIs are high $\delta^{34}S$ values become lighter which is the characteristic of the TMI catalyzed pathway.

The reason that we don't show the same correlation for larger sizes is the lack of data.

Caption figure 6, As the particles become larger $\delta34S$ become more negative...second "become" should be becomes. Comma should appear after "larger"

The change has been made as requested.

Figure 5. $\delta^{34}S$ ranges for F<0.49µm, E (0.49<D<0.95 µm), D (0.95<D<1.5 µm), C (1.5<D<3.0 µm), B (3.0<D<7.2 µm) and A>7.2 µm size ranges during CIMS-OFF periods. As the particles become larger, $\delta^{34}S$ becomes more negative.

Figure 8: In figure 7, the Fe concentrations go up to 300 ng/m3 and Mn to 6 ng/m3. In figure 8, the largest values are 72 and 1.2 ng/m3, respectively. The reason is unclear to me. The reason is that for the plots sulfate concentration vs Fe and Mn concentrations all data are used but for the plot $\delta^{34}S$ vs the concentration of Fe and Mn there are only a few $\delta^{34}S$ values for CIMS-OFF periods. Other $\delta^{34}S$ values were for CIMS-ON periods which we couldn't use them.

Page 16, line 3: A negative correlation was observed between F(s) and this (integrated) proxy during the daytime (r=0.72, P-value<0.05).

This is quite remarkable (and not plotted). If any, a positive correlation would be expected. Anyhow, is seems correlation appear easy, and alternative explanations for some of the plotted correlations may be possible. Plotting only the relations that fit the story is questionable. On page 19, it is concluded that: "Negative correlation between F(s) and integrated OH proxy indicates the minor importance of OH oxidation pathway during daytime." I generally agree, but this provides no explanation for the negative correlation.

An explanation for the negative correlation has been added to the paper as well as the plot of F(s) vs the OH production rate.

Page 12, line 4: A negative correlation was observed between F(s) and this (integrated) OH production rate during the daytime (r = -0.72, P-value<0.05) (Figure A2).

However, two data points with the highest RH (25 and 26 August) drive this correlation, and no correlation was observed for the remainder of the samples. This suggests that there may be $SO_2$ oxidation pathways in addition to OH during the day in this region.

Page 22, top: "Since indicators of soil dust (Ca and Mg) were not present on F samples the decrease in $\delta34S$ values cannot be associated with primary sulfate from haul road dust and soil

in this size fraction (Phillips-Smith et al., 2017). Instead, the lighter δ34S for F samples indicates oxidation and TMI is the only fractionation pathway known for which this is feasible. ". Here I am lost. In figure 6, I see that the F-samples are the most enriched in δ34S, somewhere between 2 and 15 per mil. So, I do not see the role of TMI here, because I compare this to the measured δ34S value of SO2, which was mentioned one page earlier to be: +7.9 ± 2.1 ‰. So, I do not see what is meant by "the lighter δ34S for F samples". For the other (A-E) samples the situation is clearer, but this discussion is very confusing.

The sulfur isotopic composition of sulfate aerosols in this size range (F) are lighter than the isotopic composition of $SO_2$ measured at the same time. This argument is not mentioned in the new version.

Section 5.3: This line of argumentation sounds relatively straight-forward, although it seems at odds with known speeds of conversion. Also, I do not understand why the Criegee mechanism is not discussed here.

Yes, what we found for the $SO_2$ oxidation rate is surprising, but it is only observed for a source which is local and close to ground. It seems that emitted enriched $SO_2$ from the CIMS is oxidized on the surface of aerosols in all size fractions that we measure. The oxidation seems to occur in aqueous phase and that's the reason we didn't talk about $SO_2$ oxidation by Criegee intermediates in this section.

Page 1, line 19: Analysis of $^{34}S$ enhancements of sulfate and $SO_2$ during CIMS-ON periods indicated rapid oxidation of $SO_2$ from this local source at ground level on the surface of aerosols before reaching the high-volume sampler or on the collected aerosols on the filters in the high-volume sampler.

**Report #2:**
Thank you for your valuable comments. Many of the recommended changes you mentioned in your report were addressed as described below.

Although the analyses and data obtained are very good, they represent a short period of time and do not reflect the variability on a seasonal scale.

 Although the data represent a short period of time and do not reflect the variability on a seasonal scale, Soares et al. (2018) showed that short-term measurements are more suitable for source identification. He mentioned that the source signals of $NO_2$ and $SO_2$ emissions are available in hourly to daily time scales and long-term observation may cause a loss in short-term variation. A reference to this paper and a sentence pointing out that this is the case has been added to the introduction section on page 5, line 2 and reads as follows: Although the data represent a short period of time and do not reflect the variability on a seasonal scale, Soares et al. (2018) showed that short term measurements are more suitable for source identification. He mentioned that the source signals of $NO_2$ and $SO_2$ emissions are available in hourly to daily time scales and long-term observation may cause a loss in short term variation.

More information about these and impact on weather/climate and the environment are needed to motivate why this study has been done.

We have expanded the introduction to include the following two paragraphs clarifying the impact on weather/climate and the environment.

Page 2, line 4: Sulfate aerosols are known to impact ecosystems and climate through their deposition and radiative effects. The deposition of sulfate aerosols can cause acidification of soils and lakes (Gerhardsson, 1994). Furthermore, their direct and indirect radiative effects can change the radiative budget at regional scales and alter climate (IPCC, 2001).

Sulfur dioxide ($SO_2$) is converted to sulfate in homogeneous and heterogeneous reactions. The oxidation pathway is a very important factor to determine the effects of the sulfate formed on the environment.

Page 2, lime 32: Gas phase oxidation of $SO_2$ by hydroxyl radicals (OH) produces sulfuric acid ($H_2SO_4$) gas, which can nucleate in the atmosphere to form new particles (Tanaka et al.,1994; Kulmala et al., 2004). These newly formed aerosol particles are buoyant and can be dispersed far from the emission source. Newly formed sulfate aerosols also impact direct radiative forcing by scattering sunlight back to space. These particles can grow by the addition of organics to create a large number of accumulation mode aerosols which are more easily deposited on local surfaces, increasing the potential for acidification at regional to local scales. They also have the ability to form cloud condensation nuclei (CCN) (Kulmala et al., 2004, 2007; Benson et al., 2008). After forming CCN they can increase the albedo and lifetime of clouds (Twomey, 1991; Boucher and Lohmann, 1995).

Page 3, line 16: Heterogeneous oxidation of $SO_2$ primarily occurs in cloud droplets, although oxidation on the surface of aerosols can be important regionally (Chin and Jacob, 1996). Heterogeneous oxidation prevents $H_2SO_4$ gas production and new particle formation. Sulfate formed by this pathway can modify the aerosol size distribution, which affects both direct and indirect aerosol forcing. Scattering efficiency of the particle population can be increased which is responsible for direct scattering (Hegg et al., 2004; Yuskiewicz et al., 1999). In addition, acidity of aerosols as well as their CCN activity of the particle population can be modified and affect the indirect radiative forcing (Mertes et al., 2005a, b).

Also, the conclusions do not show what are the new findings by the work that differ from earlier investigations.

Addressing this concern about the new findings we added the following paragraphs to the introduction and conclusion sections of the manuscript.

Page 4, line 6: Field studies suggested that TMI-catalyzed oxidation is the dominant sulfate formation pathway in polluted environments in winter (Jacob et al., 1984, 1989; Jacob and Hoffmann, 1983). Oxygen isotope measurements of sulfate aerosols collected at Alert, Canada (82.5°N, 62.3°W) showed that TMI-catalyzed $SO_2$ oxidation is significant during winter (McCabe et al., 2006). Recent studies have shown that the TMI-catalyzed oxidation pathway is underestimated (more than an order of magnitude) in all current atmospheric chemistry models (Harris et al., 2013a, b). For example, Harris et al. (2013a) measured the sulfur isotopic composition of $SO_2$ upwind and downwind of clouds and used the difference to calculate the fractionation that occurred for in-cloud $SO_2$ oxidation. They showed that $SO_2$ oxidation catalyzed by natural TMIs on mineral dust is the dominant in-cloud oxidation pathway and is underestimated by more than an order of magnitude in current atmospheric models. To the best of our knowledge, there is no study to investigate the importance of the TMI-catalyzed pathway

in $SO_2$ oxidation on the surface of aerosols in highly polluted areas such as the Alberta oil sands region during summer.

[revised manuscript text omitted]

---

## Author Response (AR2)

FACULTY OF SCIENCE
Department of Physics & Astronomy
The University of Calgary
Calgary, Alberta
T2N 1N4 Canada

Telephone:  (403) 220-5405
Fax: (403) 210-7773
Email: alnorman@ucalgary.ca

May 10, 2018

acp-2017-1023

Re: Response to Editor's comments

Dear Shao-Meng,

Thanks very much for your valuable feedback on our manuscript on $SO_2$ oxidation as part of the 2013 JOSM summer field campaign 'Stable sulfur isotope measurements to trace the fate of $SO_2$ in the Athabasca oil sands region".  Co-authors Hans Osthoff, Travis Tokarek and I agreed with your statement that "definitive statements are made where cautions may be exercised".  We worked together to fine tune the wording to more appropriately reflect what is warranted based on the data (e.g. points 4, 20 below).  I have responded to each of your points (blue text):

1.  For all the discussions on the oxidation pathways by OH, $H_2O_2$, TMI, criegee biradicals, the presentation and discussions have an underlying assumption, and that is, they all occur in the same air parcel.  One can argue that the time zero of the air parcel is when all chemical ingredients are mixed.  In reality, a well mixed plume with $SO_2$, TMI, and the biradicals all included is not easy to achieve on the spatial scale of 10 km, or the approximate distance from the stacks to the AMS13 site.  We already know that the $SO_2$ plume was emitted at height, and the TMI and biradicals are associated with ground sources.  One wonders about the time scale before they mix, and the degree of conversion of $SO_2$ to sulfate in that time span.

Throughout the text I have attempted to make it clear that we do not assume that all oxidation pathways occur during transport to the site.  $SO_2$ oxidation by OH, $H_2O_2$, and $O_3$ catalyzed TMI oxidation assumed to occur aloft during pollutant transport whereas once the $SO_2$ mixes downward at the AMS13 site it encounters higher concentrations of Criegee radicals and is oxidized rapidly.  I have added several clauses throughout the text (itemized below) to be explicit that rapid oxidation is expected close to the surface.

2.  TMI should occur mostly in the coarse particles given the evidence that they are from dust and upgrader sources; yet you are comparing their concentrations with F(s) for the very small particles.

Fe and Mn that correlate with our F(s) values are for $PM_{2.5}$ (Phillips-Smith et al., 2017) (see points 5, 13, 14, 15 etc. below). Proemse et al., used $\delta^{18}O$ values as well as measurements from the stacks to demonstrate that $PM_{2.5}$ in the region is dominated by (>90%) secondary, rather than primary sulfate. Although we expected TMI to occur mainly in the coarse particles, the very light isotope values for larger size aerosols suggest it IS important in that size range as well, we also saw this for aerosols <2.0 microns in diameter and had a much fuller data set to work with as some measurements for large size aerosols were below detection limits.

3.  F(s) can be affected by non-chemical processes, in particular dry deposition processes. The mixing process of the elevated $SO_2$ plume with ground plumes of TMI/biradicals is certainly conducive to preferential dry deposition of $SO_2$ compared to fine particle sulfate, thus increasing the value of F(s) without any chemistry.

While F(s) could potentially be affected by the dry deposition process you suggest our isotope data, and the $^{34}S$ enriched S in particular, are consistent with chemical rather than physical processes driving F(s). A sentence has been added at the bottom of page 9 (lines 29-31) that states variations between $SO_2$ and sulfate dry deposition velocities are neglected (see point 2 below).

4.  Our MCM modelling for the lagrangian experiments from aircraft during our transformation flights shows that criegee biradicals contributed little to $SO_2$ conversion in clear air even using the latest kinetics results for the biradical reactions with $SO_2$, most of the conversion still dominated by the OH. Granted that you have no access to this information, but I am just wondering that the correlations that you saw in F(s) versus the concentrations of the terpenes can be explained by air mass passing over the site, rather than a pure chemical explanation.

The very surprising result that is strongly supported by our isotope data, both the tracer and non-tracer results, is that it demonstrates that the driver of F(s) is chemical conversion of $SO_2$ rather than physical processes at ground level. The discrepancies between what were observed aloft and at ground level may well be due to the very high concentrations of Criegee biradicals nearer the canopy. The situation may be that $SO_2$ oxidation at ground level is enhanced relative to the air above.

5.  I guess my issue with the presentation is that all the correlations you have discovered are presented from a purely chemical process perspective. While chemical processes probably play a (important) role, they are probably not the only explanations.

There are no physical processes that can explain the isotope fractionation observed. Instead, the data point to TMI catalysis: potentially this occurs aloft as Fe and Mn from the upgrader in aerosols <$PM_{2.5}$ is transported with $SO_2$, followed by Criegee biradical oxidation once the plume descends close to the ground.

Sincerely,

Ann-Lise Norman

Professor, Physics & Astronomy and the

Environmental Science Program
The University of Calgary

Changes in the manuscript are highlighted in the pdf accompanying this letter. The rationale for each change (except those that are grammatical in nature) is provided below.

1. Abstract – lines 15-24: The order in which sentences were presented has been reorganized. The word "indicate" has been changed to "suggests", and the phrase "highly polluted regions" has been changed to "in the study region".

2. 4.3.2. S Conversion Ratio – page 9, lines 28-31: Additional sentences were added to clearly state that dry deposition was not considered in this study.

3. 5.1. Results - page 11, lines 5-7: Wording was clarified to better convey the intent. The phrase "since $F_{<0.49um}$ fraction contains particles even in smaller sizes than 2.5 $\mu$m" has been removed. The word "this" in the following sentence has been changed to "<0.49 $\mu$m size range". Note that this section is not highlighted as text was removed.

4. 5.1. Results - page 13, line 3: Wording clarified: the phrase "is believed to be" was added in front of "the dominant $SO_2$ transformation pathway" to be less definitive.

5. 5.1. Results – page 13, line 6: The words "in $PM_{2.5}$" have been added for clarification (note this text is not highlighted in the pdf).

6. 5.2 Results – page 14, line 27: Sentence removed. "All the available data for blank corrected size fractions A and B were negative."

7. 5.2.1 Results – page 16, line 1: Clarification of how these size fractions were measured. Added the word "impactor".

8. 5.2.1. Results – page 16, line 11: Added citation.

9. 5.3. Results – page 17, line 4: the word "very" was removed before "unexpected" (not highlighted).

10. 5.4. Results – page 20, line 4: Reworded the text to be less definitive.

11. 5.4. Results – page 20, line 6: Added a citation.

12. 6.1. Discussion – page 21, line 5: Clarified the reworded the sentence and deleted a second sentence that was unnecessary.

13. 6.1. Discussion – page 21, lines 9&10: Added a citation and changed the word "indicates" to "suggests".

14. 6.1. Discussion – page 21, line 14: Removed the word "consistently".

15. 6.1. Discussion – page 22, line 27: Changed the phrase "showed evidence for" to "suggests".

16. 6.1. Discussion – page 22, line 33: Reworded this sentence to clarify the meaning.

17. 6.1. Discussion – page 23, lines 3&4: Clarified the intent of this sentence.

18. 6.2. Discussion – page 23, line 28: Added the phrase "at ground level".

19. 6.3. Discussion – page 24, lines 2&3: Rephrased to clarify the intent.

20. 6.3. Discussion – page 24, lines 5&6: Added the phrase "close to the surface" to clarify that we recognize the importance of Criegee radicals close to the ground rather than aloft.

21. 6.3. Discussion – page 24, line 16. A sentence was removed. "Styrene reacts predominantly with OH but can undergo ozonolysis to form Criegee biradicals".

22. 7. Conclusions – page 24, line 21. This sentence was moved forward and refers to AMS 13 rather than a general statement.

23. 7. Conclusions – page 24, line 26. Added the word "rapidly", broke sentence in two.

24. 7. Conclusions – page 24, lines 31 to 33. Emphasized the rapid oxidation of $SO_2$ and reworded the concluding sentence to better reflect that our results were consistent with recent literature.

[revised manuscript text omitted]